# DIFFERENTIABLE INTEGER LINEAR PROGRAMMING

**Zijie Geng[1,2]\*, Jie Wang[1,2]†, Xijun Li[3], Fangzhou Zhu[4], Jianye Hao[4,5], Bin Li[1], Feng Wu[1,2]**
[1] University of Science and Technology of China
[2] MoE Key Laboratory of Brain-inspired Intelligent Perception and Cognition
[3] Shanghai Jiao Tong University    [4] Noah's Ark Lab, Huawei    [5] Tianjin University
`zijiegeng@mail.ustc.edu.cn`, `{jiewangx,binli,fengwu}@ustc.edu.cn`
`lixijun@sjtu.edu.cn`, `{zhufangzhou,haojianye}@huawei.com`

## ABSTRACT

Machine learning (ML) techniques have shown great potential in generating high-quality solutions for integer linear programs (ILPs). However, existing methods typically rely on a *supervised learning* paradigm, leading to (1) *expensive training cost* due to repeated invocations of traditional solvers to generate training labels, and (2) *plausible yet infeasible solutions* due to the misalignment between the training objective (minimizing prediction loss) and the inference objective (generating high-quality solutions). To tackle this challenge, we propose **DiffILO** (**Diff**erentiable **I**nteger **L**inear Programming **O**ptimization), an *unsupervised learning paradigm for learning to solve ILPs*. Specifically, through a novel probabilistic modeling, DiffILO reformulates ILPs—discrete and constrained optimization problems—into continuous, differentiable (almost everywhere), and unconstrained optimization problems. This reformulation enables DiffILO to simultaneously solve ILPs and train the model via straightforward gradient descent, providing two major advantages. First, it significantly reduces the training cost, as the training process does not need the aid of traditional solvers at all. Second, it facilitates the generation of feasible and high-quality solutions, as the model *learns to solve ILPs* in an end-to-end manner, thus aligning the training and inference objectives. Experiments on commonly used ILP datasets demonstrate that DiffILO not only achieves an average training speedup of 13.2 times compared to supervised methods, but also outperforms them by generating heuristic solutions with significantly higher feasibility ratios and much better solution qualities.

## 1 INTRODUCTION

Integer linear programs (ILPs) are powerful tools in various areas such as operations research, mathematics, and engineering (Bixby et al., 2004; Bengio et al., 2021). They are able to model a broad range of combinatorial optimization (CO) problems and find diverse real-world applications, including scheduling (Ryan & Foster, 1981), planning (Beyer et al., 2016), and network design (Koster et al., 2010). Despite their significant importance, ILPs inherently exhibit a complex combinatorial nature and are known as quintessential $\mathcal{NP}$-hard problems, posing substantial challenges in solving them efficiently. Therefore, extensive research and development efforts have been dedicated to advancing ILP solvers, such as SCIP (Achterberg, 2009) and Gurobi (Gurobi Optimization, 2021). These solvers are mainly based on traditional algorithms such as Branch-and-Bound (B&B) (Land & Doig, 2010) and Branch-and-Cut (B&C) (Mitchell, 2002), which are meticulously enhanced with various heuristics to improve efficiency and accuracy.

Machine learning (ML) techniques, especially deep neural networks (DNNs), have recently shown great potential in solving or aiding the resolution of ILPs (Zhang et al., 2023; Li et al., 2024a). In practice, ILPs within specific scenarios often exhibit similar structures or patterns across instances, enabling ML methods to automatically identify and exploit these patterns to reduce computational complexity in a data-driven manner (Gasse et al., 2019). When trained on a dataset of instances,

---

\*This work was done when Zijie Geng was an intern at Huawei.
†Corresponding author.

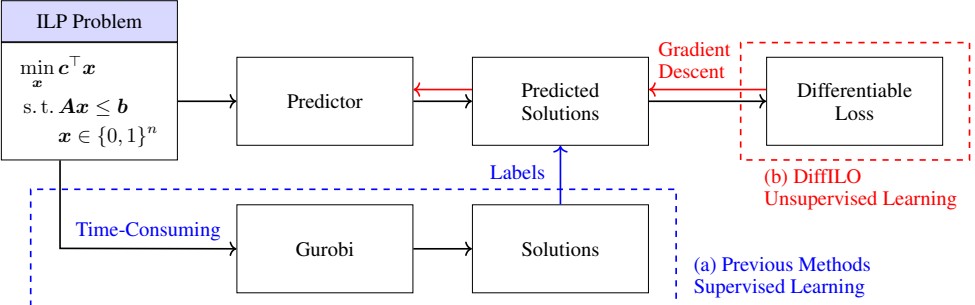

Figure 1: **(a) Previous works mainly use supervised learning.** They employ Gurobi to obtain solutions that serve as training labels, which is time-consuming. **(b) Our proposed DiffILO is an unsupervised learning approach.** Its key idea is to design a differentiable loss function, enabling straightforward gradient descent methods to optimize the problems and the predictor simultaneously.

DNNs are able to generalize to novel but similar instances, making decisions according to past experiences in a short time frame. The growing intersection of CO and ML has attracted considerable attention due to its potential to drive innovation and offer mutual benefits to both fields. Research efforts in this area can be generally categorized into two main streams. Some studies integrate ML into the traditional branch-and-bound framework, enhancing components such as branching decisions (Gasse et al., 2019; Kuang et al., 2024a;b; Liu et al., 2023; 2024b), separation processes (Li et al., 2023a), presolving (Kuang et al., 2023), and cut selection (Wang et al., 2023c; Ling et al., 2024; Wang et al., 2024b). Another line of research, which is the focus of this paper, directly employs ML to predict solutions (Khalil et al., 2022; Liu et al., 2024a; 2025). Recent advancements, such as Neural Diving (Nair et al., 2020; Yoon, 2022) and Predict-and-Search (Han et al., 2023; Huang et al., 2024; Zeng et al., 2024), have shown effectiveness in reducing the solving time. These methods typically follow a *supervised learning* paradigm, using traditional solvers like SCIP (Achterberg, 2009) and Gurobi (Gurobi Optimization, 2021) to generate near-optimal solutions as training labels. A predictor is trained to predict the near-optimal solutions from the given instances. Sophisticated heuristics are then applied to exploit the predicted solutions, thus accelerating the solving process.

Despite these achievements, supervised learning methods still present significant challenges. First, they are time-extensive due to the need to repeatedly invoke traditional solvers to generate training labels. This is a fundamental limitation of supervised learning approaches (Wang et al., 2024a), which has driven researchers across various fields to turn their attention towards *unsupervised learning* approaches (Wang et al., 2022). Second, there exists an inherent misalignment between the objectives of minimizing prediction errors during training and generating high-quality solutions during inference. In other words, the predictor is trained to mimic the provided solutions rather than independently solving the problems. Consequently, these methods often yield plausible yet infeasible solutions (Zeng et al., 2024). In light of these issues, *developing a differentiable approach for solving ILPs in an unsupervised learning* paradigm becomes especially attractive. As illustrated in Figure 1, the core concept involves designing a differentiable loss function to directly optimize the problems and the predictor using gradient descent methods. In the field of CO, this differentiable loss is intuitively expected to align with the optimization objectives, thus enabling an end-to-end training framework that facilitates producing high-quality solutions. From a fundamental research perspective, such a method aligns with the ongoing academic pursuit of unsupervised learning. From a practical perspective, this method promises significant advantages, significantly reducing training time while improving the quality of generated solutions (Chen et al., 2023).

To tackle this challenge, we propose **DiffILO** (**Diff**erentiable **I**nteger **L**inear Programming **O**ptimization), a novel *unsupervised learning* approach for learning to generate high-quality ILP solutions. To the best of our knowledge, DiffILO is the first method to employ pure ML techniques for training, without relying on traditional solvers, representing a front-line exploration of ML applications in the field of combinatorial optimization. Specifically, DiffILO first relaxes the binary variables into continuous ones via probabilistic modeling, where the constraints are transformed into the form of expected violation. We theoretically demonstrate the equivalence preserved in this transformation. DiffILO then adopts the penalty function method to convert the constrained problems into unconstrained ones. By further leveraging a reparameterization trick to flow the gra-

dient back-propagation, thus forming a differentiable (almost everywhere) loss function, DiffILO optimizes both the ILP problems and the predictor parameters simultaneously via straightforward gradient descent methods. The training process is entirely unsupervised, thus significantly reducing the training time by bypassing the collection of labeled data. Moreover, the end-to-end optimization aligns the objectives of training and inference, thus consistently producing feasible and high-quality solutions. Extensive experiments demonstrate that DiffILO not only achieves an average training speedup of 13.2 times compared to supervised methods, but also outperforms them by generating heuristic solutions with much higher feasibility ratios and significantly improved objective values.

## 2 RELATED WORK

### 2.1 MACHINE LEARNING FOR INTEGER LINEAR PROGRAMS

Machine learning (ML) techniques have become increasingly prevalent in addressing combinatorial optimization (CO) problems, especially integer linear programs (ILPs) and mixed-integer linear programs (MILPs) (Bengio et al., 2021; Li et al., 2023b; Zhang et al., 2023; Huawei, 2021). Some studies incorporate ML models into heuristic components in modern solvers (He et al., 2014; Baltean-Lugojan et al., 2019; Kuang et al., 2023; Li et al., 2024a), such as branching (Gasse et al., 2019), separation (Li et al., 2023a), and cut selection (Wang et al., 2023c), etc. Another line of research, which is the focus of this paper, employs ML to generate heuristic solutions (Nair et al., 2020; Yoon, 2022; Khalil et al., 2022; Ye et al., 2023b; Zeng et al., 2024). A notable recent advancement is the predict-and-search (PS) framework (Han et al., 2023; Huang et al., 2024), which first predicts initial solutions, and then employs solvers like Gurobi (Gurobi Optimization, 2021) or SCIP (Achterberg, 2009) to search within a strategically designed trust region for solution improvement. Such research is similar to the decision-focused learning (DFL) or predict-then-optimize framework (Elmachtoub & Grigas, 2022; Ferber et al., 2020; Zharmagambetov et al., 2024), which learns a model to map observable features into latent representation (e.g. coefficients in LP objective) used by solvers.

### 2.2 DIFFERENTIABLE APPROACHES

Differentiable approaches aim to construct a differentiable optimization objectives, facilitating the straightforward application of gradient descent methods in an unsupervised, end-to-end manner. These methods have been effectively implemented in fields such as Partial Differential Equation (PDE) (Holl et al., 2020; Belbute-Peres et al., 2020) and Density Functional Theory (DFT) (Kvaal et al., 2014; Mathiasen et al., 2024; Li et al., 2024b), showcasing their effectiveness in solving such continuous problems. Applying these techniques to CO problems poses challenges due to the discrete nature of these problems. In the field of CO, differentiable approaches have been tailored for some specific problems, such as Boolean Satisfiability Problem (SAT) (Amizadeh et al., 2018) and Traveling Salesman Problem (TSP) (Gaile et al., 2022). Notably, Karalias & Loukas (2020) explored differentiable approaches for combinatorial optimization on graphs and developed Erdős Goes Neural, a differentiable and unsupervised learning framework that is similar to our approach. This concept has been further explored by some following works (Wang et al., 2022; Schuetz et al., 2022; Wang & Li, 2023). However, these methods are tailored for some specific problems and depend on custom-designed differentiable loss functions specific to these cases. To the best of our knowledge, extending these techniques to general ILPs remains non-trivial.

## 3 METHODOLOGY

This section introduces our proposed DiffILO framework, with an overview depicted in Figure 2. In Section 3.1, we present the probabilistic approach that reformulates a discrete, constrained ILP problem into a continuous, unconstrained optimization problem. Next, in Section 3.2, we adopt the Gumbel Softmax technique for reparameterization, which makes the problem differentiable almost everywhere (a.e.) and facilitates an efficient resolution via straightforward stochastic gradient descent. Finally, Section 3.3 details the implementation of the DiffILO model, including its training and inference processes. The proofs and additional implementation specifics are available in Appendix A and Appendix B, respectively.

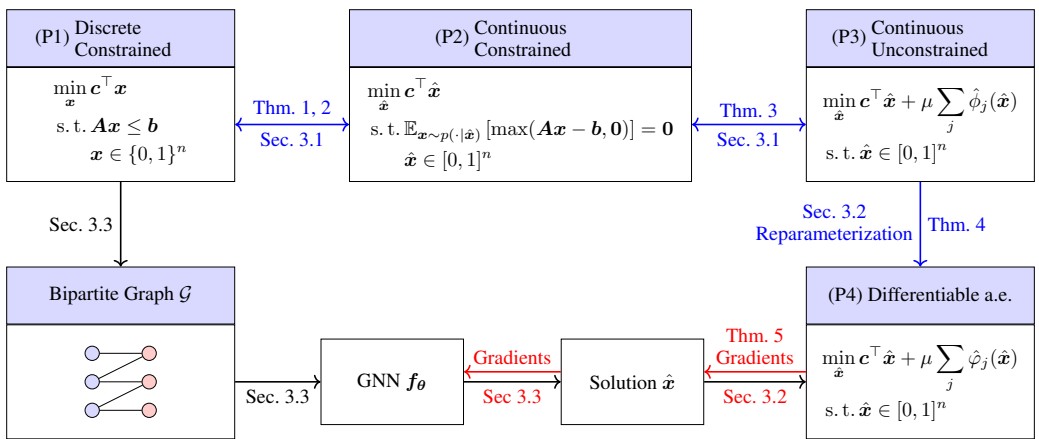

Figure 2: **Method overview of DiffILO.** We first transform the primal discrete and constrained problem into a continuous, unconstrained, and differentiable (a.e.) problem, as depicted by the blue arrows. DiffILO employs a graph neural network (GNN) to predict solutions, as depicted by the black arrows. It optimizes both the ILP problem and the GNN parameters simultaneously through gradient descent, as depicted by the red arrows.

## 3.1 PROBABILISTIC MERIT FUNCTION

We focus on integer linear programs (ILPs) that take the form of:

$$\min_{\boldsymbol{x}} \left\{ \boldsymbol{c}^\top \boldsymbol{x} \mid \boldsymbol{A}\boldsymbol{x} \leq \boldsymbol{b}, \boldsymbol{x} \in \{0,1\}^n \right\}. \tag{P1}$$

where $n$ denotes the number of variables, $\boldsymbol{c} \in \mathbb{R}^n$ denotes the objective coefficients, the matrix $\boldsymbol{A} = (\boldsymbol{a}_1, \boldsymbol{a}_2, \cdots, \boldsymbol{a}_m)^\top \in \mathbb{R}^{m \times n}$ denotes the constraint coefficient matrix, $m$ denotes the number of constraints, and $\boldsymbol{b} \in \mathbb{R}^m$ denotes the right-hand-side biases of the constraints.

**Remark 1.** *Without loss of generality, we focus on ILPs with binary integer variables, also referred to as 0-1 programs. This simplification is reasonable given the fact that any bounded integer program can be converted into a binary (0-1) form (Dantzig, 2016; Ye et al., 2023a).*

An ILP is quintessentially a discrete and constrained optimization problem. Our first step is to reformulate it into a continuous problem. An intuitive approach is to relax the binary variables $\boldsymbol{x} \in \{0,1\}^n$ into continuous variables $\hat{\boldsymbol{x}} \in [0,1]^n$, which is commonly known as the linear programming (LP) relaxation (Agmon, 1954; Bixby et al., 2004). However, clearly such relaxation alters the original solution space. To achieve an equivalent reformulation, we adopt a probabilistic approach (Karalias & Loukas, 2020), interpreting the continuous variables $\hat{\boldsymbol{x}}$ as probabilities associated with the binary variables. Specifically, each binary variable $x_i$ is assumed to follow a Bernoulli distribution, writing $x_i \sim \text{Bernoulli}(\hat{x}_i)$. The distribution of $\boldsymbol{x}$ is denoted as $\boldsymbol{x} \sim p(\cdot|\hat{\boldsymbol{x}})$. Based on the probabilistic modeling, we reformulate the primal problem (P1) into:

$$\min_{\hat{\boldsymbol{x}}} \left\{ \boldsymbol{c}^\top \hat{\boldsymbol{x}} \mid \mathbb{E}_{\boldsymbol{x} \sim p(\cdot|\hat{\boldsymbol{x}})}[\max(\boldsymbol{A}\boldsymbol{x} - \boldsymbol{b}, \boldsymbol{0})] = \boldsymbol{0}, \hat{\boldsymbol{x}} \in [0,1]^n \right\}. \tag{P2}$$

**Remark 2.** *The novel reformulation in (P2) is motivated by the intuition that minimizing the expected constraint violations can restrict the distribution's support to the set of optimal solutions. For example, if a component $\hat{x}_i \in (0,1)$, then the constraint $\mathbb{E}_{x \sim \text{Bernoulli}(\hat{x}_i)}[\max(ax - b, 0)] = 0$ will ensure that $ax \leq b$ holds for both $x = 0$ and $x = 1$. Otherwise, if $\hat{x}_i = 0$ or $\hat{x}_i = 1$, $\text{Bernoulli}(\hat{x}_i)$ becomes a deterministic distribution and it indicates that $a\hat{x}_i \leq b$ holds.*

We now present the fundamental theoretical properties of (P2), establishing its equivalence with the primal problem (P1). Theorem 1 demonstrates the equivalence betweeen the two problems in terms of feasibility and solvability. Then, Theorem 2 shows that, loosely speaking, the optimal solutions to (P2) coincide with the optimal solutions to (P1).

**Theorem 1.** *The problem (P2) is feasible (and solvable, i.e., it admits at least one optimal solution) if and only if (P1) is feasible (and solvable).*

**Theorem 2.** *Let $\mathcal{I}_{\boldsymbol{c}} \triangleq \{i \in [n] : c_i \neq 0\}$. Then the following statements hold:*

1. *Suppose $\boldsymbol{x}^* \in \{0, 1\}^n$ is an optimal solution to (P1). Then $\boldsymbol{x}^*$ is also an optimal solution to (P2). If a vector $\hat{\boldsymbol{x}}^* \in [0, 1]^n$ is a feasible solution to (P2) and satisfies $\hat{x}_i^* = x_i^*$ for all $i \in \mathcal{I}_{\boldsymbol{c}}$, then $\hat{\boldsymbol{x}}^*$ is an optimal solution to (P2).*

2. *Suppose $\hat{\boldsymbol{x}}^* \in [0, 1]^n$ is an optimal solution to (P2). Then we have $\hat{x}_i^* \in \{0, 1\}$ for all $i \in \mathcal{I}_{\boldsymbol{c}}$. Let $\mathcal{I}_{\hat{\boldsymbol{x}}^*} = \{i \in [n] : \hat{x}_i^* \in \{0, 1\}\}$. If a vector $\boldsymbol{x}^* \in \{0, 1\}^n$ satisfies $x_i^* = \hat{x}_i^*$ for all $i \in \mathcal{I}_{\hat{\boldsymbol{x}}^*}$, then $\boldsymbol{x}^*$ is an optimal solution to (P1).*

We can now conclude that transforming (P1) into its continuous format (P2) is well-justified. The next step is to apply the penalty function method to recast the constrained problem (P2) into an unconstrained format. We define $\phi_j(\boldsymbol{x}) \triangleq \max(\boldsymbol{a}_j^\top \boldsymbol{x} - b_j, 0)$ and $\hat{\phi}_j(\hat{\boldsymbol{x}}) \triangleq \mathbb{E}_{\boldsymbol{x} \sim p(\cdot|\hat{\boldsymbol{x}})}[\phi_j(\boldsymbol{x})]$ for each constraint indexed by $j \in [m]$. Plugging these penalty functions into the optimization objective results in the following unconstrained problem:

$$\min_{\hat{\boldsymbol{x}}}\{F_\mu(\hat{\boldsymbol{x}}) \triangleq \boldsymbol{c}^\top \hat{\boldsymbol{x}} + \mu \sum_j \hat{\phi}_j(\hat{\boldsymbol{x}}) \mid \hat{\boldsymbol{x}} \in [0, 1]^n\}. \tag{P3}$$

Here, the function $F_\mu(\hat{\boldsymbol{x}})$ is referred to as a merit function (Nocedal & Wright, 1999). Its exactness is supported by the following theorem, which states that for a sufficiently large penalty coefficient $\mu$, the penalty function method preserves the optimal solutions of the original problem.

**Theorem 3.** *There exists a positive scalar $\mu^* > 0$ such that for any $\mu > \mu^*$, any optimal solution to (P3) is also an optimal solution to (P2).*

**Remark 3.** *The conclusion in Theorem 3 differs from the established exact penalty function theory, which typically assumes the existence of a first-order Karush-Kuhn-Tucker (KKT) point (Di Pillo & Grippo, 1989). In contrast, our proof leverages the combinatorial properties of the primal problem.*

**Remark 4.** *The probabilistic modeling and penalty function approach have been used in some previous studies for combinatorial optimization problems (Karalias & Loukas, 2020; Wang et al., 2022). However, these methods depend on predefined closed-form constraint penalties, which are hard to derive for general ILPs. As a result, they are applicable to some specific problems rather than general ILPs. Our key technical innovation lies in the transformation of constraints into an expectation form in (P2), eliminating the need for closed-form penalty designs.*

## 3.2 GRADIENT BACK-PROPAGATION FOR OPTIMIZATION

We are now interested in how to apply gradient-based methods, such as Stochastic Gradient Descent (SGD), to optimize (P3). This involves estimating the gradients $\nabla_{\hat{\boldsymbol{x}}} \hat{\phi}_j(\hat{\boldsymbol{x}}) = \nabla_{\hat{\boldsymbol{x}}} \mathbb{E}_{\boldsymbol{x} \sim p(\cdot|\hat{\boldsymbol{x}})}[\phi_j(\boldsymbol{x})]$. Since $\hat{\boldsymbol{x}}$ appears in the sampling distribution and it is hard to derive a closed-form expression for the expectation in general ILPs, estimating the gradients is non-trivial. To address this, we employ the reparameterization trick to enable accurate and low-variance gradient estimates, thus facilitating efficient gradient back-propagation.

**Remark 5.** *An alternative approach for handling such non-differentiable computation graphs involving sampling is REINFORCE (also known as the score function estimator) (Williams, 1992). However, REINFORCE falls short as it does not explicitly propagate gradients from $\phi_j(\boldsymbol{x})$, leading to a potentially less efficient optimization process (Jang et al., 2017). Therefore, we favor the reparameterization trick in our approach. See Appendix D.2 for more details.*

We adopt the relaxed Bernoulli distribution (Maddison et al., 2016; Wang & Yin, 2020) for reparameterization, which is based on a simple observation illustrated in the following lemma. We denote the sigmoid function as $\sigma(z) \triangleq \frac{1}{1+e^{-z}}$, and its inverse, known as the logit function, as $\tau(p) \triangleq \sigma^{(-1)}(p) = \log(\frac{p}{1-p})$. We also denote the uniform distribution over $(0, 1)$ as $\mathcal{U}(0, 1)$.

**Lemma 1.** *(Restated from (Maddison et al., 2016)) Let $\hat{x} \in (0, 1)$ and $\epsilon$ be a random variable sampled from $\mathcal{U}(0, 1)$. We define $\xi(\hat{x}; \epsilon) \triangleq \sigma(\tau(\hat{x}) + \tau(\epsilon))$. It follows that $P(\xi(\hat{x}; \epsilon) > 0.5) = \hat{x}$.*

**Remark 6.** *The distribution of $\xi(\hat{x}; \epsilon)$ is the so-called relaxed Bernoulli distribution. It serves as a "soft" approximation of the discrete Bernoulli distribution, enabling the gradient flow during back-propagation. It is also a specific application of the Gumbel-Softmax trick (Jang et al., 2017), and is also relevant to the random perturbed optimizers (Berthet et al., 2020).*

Building on Lemma 1, we present the following theorem for reparameterization.

**Theorem 4.** *Let $\hat{\boldsymbol{x}} = (\hat{x}_1, \cdots, \hat{x}_n) \in (0,1)^n$, and $\boldsymbol{\epsilon} = (\epsilon_1, \cdots, \epsilon_n)^\top$ be a random vector, where each $\epsilon_i$ is independently and identically distributed (i.i.d.) as $\epsilon_i \sim \mathcal{U}(0,1)$, writing $\boldsymbol{\epsilon} \sim p_{\boldsymbol{\epsilon}}(\cdot)$. Let $\boldsymbol{\xi}(\hat{\boldsymbol{x}}; \boldsymbol{\epsilon}) \triangleq (\xi_1, \cdots, \xi_n)^\top$, where $\xi_i = \xi(\hat{x}_i; \epsilon_i)$ is defined as in Lemma 1. Let $\boldsymbol{\psi}(\hat{\boldsymbol{x}}; \boldsymbol{\epsilon}) \triangleq (\psi_1, \cdots, \psi_n)^\top$, where $\psi_i = [\xi_i]$ is the binary rounded value of $\xi_i$. It follows that:*

$$\hat{\phi}_j(\hat{\boldsymbol{x}}) = \mathbb{E}_{\boldsymbol{x} \sim p(\cdot|\hat{\boldsymbol{x}})} [\phi_j(\boldsymbol{x})] = \mathbb{E}_{\boldsymbol{\epsilon} \sim p_{\boldsymbol{\epsilon}}(\cdot)} [\phi_j(\boldsymbol{\psi}(\hat{\boldsymbol{x}}; \boldsymbol{\epsilon}))]. \tag{1}$$

Theorem 4 validates the effectiveness of the relaxed Bernoulli to reparameterization, in the sense that, the term $\hat{\phi}_j(\hat{\boldsymbol{x}})$ can be accurately calculated by sampling random variables $\boldsymbol{\epsilon}$ from a non-parametric distribution $p_{\boldsymbol{\epsilon}}(\cdot)$. However, note that the gradients derived from $\boldsymbol{\psi}$, due to the existence of the rounding function, vanish everywhere. Therefore, while $\boldsymbol{\psi}$ can be used to compute the values of $\phi_j$, we use $\boldsymbol{\xi}$ to flow gradients from $\phi_j$ to $\hat{\boldsymbol{x}}$. Formally, we observe:

$$\begin{aligned} \phi_j(\boldsymbol{\psi}(\hat{\boldsymbol{x}}; \boldsymbol{\epsilon})) &= \max\{\boldsymbol{a}_j^\top \boldsymbol{\psi}(\hat{\boldsymbol{x}}; \boldsymbol{\epsilon}) - b_j, 0\} \\ &= (\boldsymbol{a}_j^\top \boldsymbol{\psi}(\hat{\boldsymbol{x}}; \boldsymbol{\epsilon}) - b_j) \mathbb{I}(\boldsymbol{a}_j^\top \boldsymbol{\psi}(\hat{\boldsymbol{x}}; \boldsymbol{\epsilon}) - b_j > 0) \\ &\approx (\boldsymbol{a}_j^\top \boldsymbol{\xi}(\hat{\boldsymbol{x}}; \boldsymbol{\epsilon}) - b_j) \mathbb{I}(\boldsymbol{a}_j^\top \boldsymbol{\psi}(\hat{\boldsymbol{x}}; \boldsymbol{\epsilon}) - b_j > 0), \end{aligned} \tag{2}$$

where $\mathbb{I}(\cdot)$ denotes the indicator function. In Equation 2, $\boldsymbol{\psi}$ is used to accurately determine whether the constraints are violated, thus preserving the combinatorial properties, while $\boldsymbol{\xi}$ acts as a surrogate to flow the gradient back-propagation. Consequently, we can approximate $\hat{\phi}_j(\hat{\boldsymbol{x}})$ as

$$\hat{\phi}_j(\hat{\boldsymbol{x}}) \approx \hat{\varphi}_j(\hat{\boldsymbol{x}}) \triangleq \mathbb{E}_{\boldsymbol{\epsilon} \sim p_{\boldsymbol{\epsilon}}(\cdot)} [\varphi_j(\hat{\boldsymbol{x}}; \boldsymbol{\epsilon})], \tag{3}$$

where

$$\varphi_j(\hat{\boldsymbol{x}}; \boldsymbol{\epsilon}) \triangleq (\boldsymbol{a}_j^\top \boldsymbol{\xi}(\hat{\boldsymbol{x}}; \boldsymbol{\epsilon}) - b_j) \mathbb{I}(\boldsymbol{a}_j^\top \boldsymbol{\psi}(\hat{\boldsymbol{x}}; \boldsymbol{\epsilon}) - b_j > 0). \tag{4}$$

The advantage of $\hat{\varphi}_j(\hat{\boldsymbol{x}})$ is that it is differentiable almost everywhere (a.e.), allowing for efficient gradient back-propagation. Therefore, we define the new surrogate problem as:

$$\min_{\hat{\boldsymbol{x}}} \{\mathcal{F}_\mu(\hat{\boldsymbol{x}}) \triangleq \boldsymbol{c}^\top \hat{\boldsymbol{x}} + \mu \sum_{j=1}^{m} \hat{\varphi}_j(\hat{\boldsymbol{x}}) \mid \hat{\boldsymbol{x}} \in [0,1]^n\}. \tag{P4}$$

The merit function $\mathcal{F}_\mu(\hat{\boldsymbol{x}})$ defined in (P4) is differentiable a.e., and thus (P4) can be resolved through gradient descent. Formally, we have the following theorem.

**Theorem 5.** *The merit function $\mathcal{F}_\mu(\hat{\boldsymbol{x}})$ defined in (P4) is differentiable almost everywhere (a.e.) in $(0,1)^n$. At the differentiable points, the gradient is given by:*

$$\nabla_{\hat{\boldsymbol{x}}} \mathcal{F}_\mu(\hat{\boldsymbol{x}}) = \boldsymbol{c} + \mu \sum_{j=1}^{m} \int_{\boldsymbol{\epsilon}: \boldsymbol{a}_j^\top \boldsymbol{\psi}(\hat{\boldsymbol{x}}; \boldsymbol{\epsilon}) - b_j > 0} \boldsymbol{a}_j \odot \left( \frac{\partial}{\partial \hat{\boldsymbol{x}}} \odot \boldsymbol{\xi}(\hat{\boldsymbol{x}}; \boldsymbol{\epsilon}) \right) p_{\boldsymbol{\epsilon}}(\boldsymbol{\epsilon}) \mathrm{d}\boldsymbol{\epsilon}, \tag{5}$$

*where $\odot$ denotes the element-wise product.*

**Remark 7.** *Though not differentiable everywhere, modern deep learning frameworks such as Py-Torch can properly handle such cases, even at non-differentiable points.*

**Remark 8.** *We consider $\hat{\boldsymbol{x}} \in (0,1)^n$ as gradient calculations are only necessary within the interior of the solution space. In practice, we output logits and apply a sigmoid function to map it to $(0,1)^n$ to represent the probabilities. As noted in Equation 4, we use the sampled binary solutions $\boldsymbol{x} \sim p(\cdot|\hat{\boldsymbol{x}})$ to compute constraint violations. Therefore, information from the exact solutions, rather than relaxed ones, is also properly carried out, preserving the combinatorial nature of the problem.*

### 3.3 MODEL IMPLEMENTATION

So far, we have transformed the original ILP problem into a continuous, unconstrained, and differentiable (a.e.) problem that can be efficiently solved using straightforward gradient descent. Building on this transformation, we introduce the model implementation of DiffILO, which simultaneously optimizes both the problem and the predictor parameters through direct gradient back-propagation, eliminating the need for labeled data. In line with the established practices in the field (Gasse et al., 2019; Han et al., 2023), we represent each ILP instance—which is formulated as (P1)—as a bipartite

graph $\mathcal{G} = (\mathcal{V} \cup \mathcal{W}, \mathcal{E})$, where $\mathcal{V}$ and $\mathcal{W}$ denote the sets of variables and constraints, respectively, and $\mathcal{E}$ denotes the edges corresponding to the coefficients. Further details on the data representation are available in Appendix B.1. The predictor $\boldsymbol{f_\theta}$, which is parameterized by $\boldsymbol{\theta}$ and assumed to be differentiable with respect to $\boldsymbol{\theta}$, outputs the predicted variable probabilities $\hat{\boldsymbol{x}} = \boldsymbol{f_\theta}(\mathcal{G})$. The model architecture is implemented as a Graph Neural Network (GNN) (Kipf & Welling, 2017; Shi et al., 2023; 2024; 2025), followed by a multilayer perceptron (MLP). The final layer applies a sigmoid function to ensure that the output probabilities $\hat{\boldsymbol{x}} \in (0, 1)^n$. More details on the model architecture are provided in Appendix B.2.

**Model Training**   During training, we update the parameters $\boldsymbol{\theta}$ by optimizing the merit function $\mathcal{F}_\mu(\boldsymbol{f_\theta}(\mathcal{G}))$, as defined in (P4), through gradient descent. Specifically, let $\mathcal{D} = \{\mathcal{G}_1, \cdots, \mathcal{G}_{|\mathcal{D}|}\}$ be a batch of ILP instances, each represented as a bipartite graph $\mathcal{G}_i$, with $n_i$ variables and $m_i$ constraints. Let $\varphi_{i,j}(\cdot)$ be the penalty function—corresponding to the previously defined $\varphi_j(\cdot)$—for the $i^{\text{th}}$ instance $\mathcal{G}_i$, i.e.,

$$\varphi_{i,j}(\hat{\boldsymbol{x}}; \boldsymbol{\epsilon}) \triangleq \left(\boldsymbol{a}_{i,j}^\top \boldsymbol{\xi}(\hat{\boldsymbol{x}}; \boldsymbol{\epsilon}) - b_{i,j}\right) \mathbb{I}\left(\boldsymbol{a}_{i,j}^\top \boldsymbol{\psi}(\hat{\boldsymbol{x}}; \boldsymbol{\epsilon}) - b_{i,j} > 0\right), \tag{6}$$

where $\boldsymbol{a}_{i,j}$ and $b_{i,j}$ denote the coefficients and the right-hand-side terms of the $j^{\text{th}}$ constraints, respectively. For each instance $\mathcal{G}_i$, we sample $K$ random vectors $\boldsymbol{\epsilon}_i^{(k)} \sim \mathcal{U}(0, 1)^{n_i}$, where $K$ is a hyperparameter. The training loss for this batch is defined as:

$$\mathcal{L}(\boldsymbol{\theta}; \mathcal{D}) \triangleq \frac{1}{|\mathcal{D}|} \sum_{i=1}^{|\mathcal{D}|} \mathcal{L}(\boldsymbol{\theta}; \mathcal{G}_i) = \frac{1}{|\mathcal{D}|} \sum_{i=1}^{|\mathcal{D}|} \left( \boldsymbol{c}^\top \boldsymbol{f_\theta}(\mathcal{G}_i) + \mu \sum_{j=1}^{m_i} \sum_{k=1}^{K} \varphi_{i,j}(\boldsymbol{f_\theta}(\mathcal{G}_i); \boldsymbol{\epsilon}_i^{(k)}) \right). \tag{7}$$

The gradient of the loss function is then given by

$$\nabla_{\boldsymbol{\theta}} \mathcal{L}(\boldsymbol{\theta}; \mathcal{D})$$

$$= \frac{1}{|\mathcal{D}|} \sum_{i=1}^{|\mathcal{D}|} \left( \nabla_{\boldsymbol{\theta}} \boldsymbol{f_\theta}(\mathcal{G}_i)^\top \left( \boldsymbol{c} + \mu \sum_{\substack{1 \leq j \leq m_i, 1 \leq k \leq K: \\ \boldsymbol{a}_{i,j}^\top \boldsymbol{\psi}(\boldsymbol{f_\theta}(\mathcal{G}_i); \boldsymbol{\epsilon}_i^{(k)}) - b_j > 0}} \boldsymbol{a}_{i,j} \odot \left( \frac{\partial}{\partial \hat{\boldsymbol{x}}} \odot \boldsymbol{\xi}\left(\boldsymbol{f_\theta}(\mathcal{G}_i); \boldsymbol{\epsilon}_i^{(k)}\right) \right) \right) \right). \tag{8}$$

We stabilize the training process through three useful techniques. First, to accommodate instances with ranging sizes and coefficient ranges, we apply a normalization to modify the loss function. Second, we find cosine annealing (Loshchilov & Hutter, 2016) to be beneficial for optimizing the learning schedule. Third, since the penalty coefficient $\mu$ is a critical hyperparameter in training, we introduce a dynamic and adaptive method for adjusting $\mu$. These training techniques are further explained in detail in Appendix C.2.

**Model Inference**   During inference, for a given instance $\mathcal{G}$, we can sample heuristic solutions from the predicted distributions $p(\cdot|\boldsymbol{f_\theta}(\mathcal{G}))$, which are immediately available. Generating heuristic solutions is particularly valuable in time-sensitive tasks or scenarios that require rapid decision-making, such as route planning and production scheduling.

Moreover, the generated heuristic solutions can be used to improve the behavior of existing ILP solvers to find high-quality solutions within a constrained time frame. DiffILO can theoretically be integrated into any framework that benefits from initialization with heuristic solutions, such as neural diving (Nair et al., 2020), large neighbourhood search (Huang et al., 2023), or Predict-and-Search (Han et al., 2023), which are complementary to our approach. In this paper, inspired by Han et al. (2023) but with further simplification, we add the following constraint to the initial problem

$$\sum_{\hat{x}_i = 0} x_i + \sum_{\hat{x}_i = 1} (1 - x_i) < \Delta, \tag{9}$$

where $\hat{\boldsymbol{x}}$ is the generated heuristic solution, and $\Delta$ is a hyperparameter. This constraint defines a trust region by limiting the number of variables—which are different from the predicted ones—to be fewer than $\Delta$. Besides, we provide $\hat{x}$ for the solver as an initial solution. Further details on the inference process can be found in Appendix C.3.

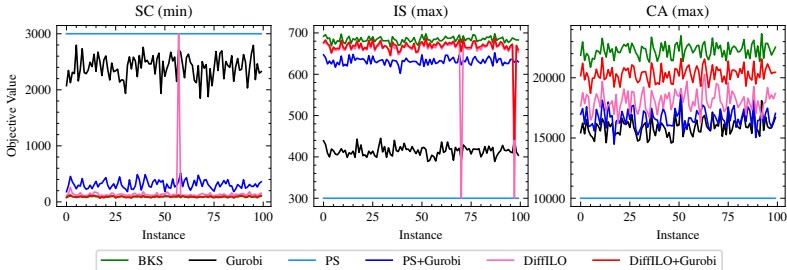

Figure 4: **The objective values of solutions generated by different approaches across the** 100 **test instances.** "BKS" denotes the best known solutions. "Gurobi" denotes the heuristic module in Gurobi that searches for heuristic solutions before the exact resolution process. For a better visualization, objective values for the instances with no feasible solution found are assigned as 3000, 300, and 10000, respectively on these datasets.

## 4 EXPERIMENTS

This section presents empirical results to demonstrate the effectiveness of our proposed DiffILO in (1) generating feasible and high-quality heuristic solutions in an end-to-end manner, and (2) improving the overall performance of traditional solvers to find high-quality solutions within a constrained time frame. We then conduct a case study to provide some additional insights into the optimization process of DiffILO. All training and evaluations are performed using the same hardware configuration, specifically an Intel(R) Xeon(R) Gold 6246R CPU @ 3.40GHz, and an NVIDIA GeForce RTX 3090 GPU. Code is available at `https://github.com/MIRALab-USTC/L2O-DiffILO`. More experimental details can be found in Appendix C.

**Benchmarks** We evaluate our method mainly on three widely used ILP problem benchmarks: set covering (SC) (Balas & Ho, 1980), maximum independent set problem (IS) (Bergman et al., 2015), and combinatorial auctions (CA) (Leyton-Brown et al., 2000). The datasets are generated using the code from Gasse et al. (2019). Following the settings described in the PS paper (Han et al., 2023), we generate 400 instances for each benchmark, 240 for training, 60 for validation, and 100 for testing, respectively. Additional details about these datasets can be found in Appendix C.1. To demonstrate the effectiveness of DiffILO on realistic datasets, we conduct experiments on two subsets of MIPLIB 2017 (Gleixner et al., 2021). More details are in Appendix D.3.

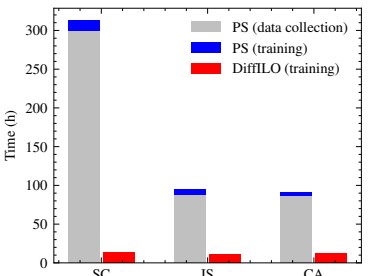

Figure 3: Training times of PS and DiffILO on different datasets. Data collection refers to the time spent on solving training and validation instances to obtain labels, while training denotes the time spent on training the neural networks.

**Baselines** We compare our method against two main categories of baselines. First, we include traditional solvers, SCIP (Achterberg, 2009) and Gurobi (Gurobi Optimization, 2021), to evaluate whether the heuristic solutions generated by DiffILO can accelerate the solving process. Second, we compare DiffILO with the Predict-and-Search (PS) framework (Han et al., 2023), which first predicts solutions and then employs SCIP or Gurobi to search within a trust region for further improvement. PS and Neural Diving (ND) (Nair et al., 2020) are both representative supervised learning methods. However, since the prediction components in PS and ND are the same, we primarily include PS as our baseline. Some more recent supervised learning approaches such as those based on contrastive learning (Huang et al., 2024) or diffusion models (Zeng et al., 2024) have not been implemented. Although these methods can enhance the performance of supervised learning, they also lead to additional training time. For fairness, DiffILO does not employ similar additional tricks either. The comparison includes capabilities of both methods to generate feasible solutions and to improve the performance of Gurobi and SCIP. We provide the results of some additional baselines, including ablation studies, in Appendix D.2.

Table 1: Average objective values obtained by different approaches at 10, 100, and 1, 000 seconds. We mark **the best values** in bold and underline the second-best values. Best known solution (BKS) refers to the solution obtained by running Gurobi for 3,600 seconds.

| | SC (min, BKS: 86.45) | | | IS (max, BKS:684.14) | | | CA (max, BKS:22272.55) | | |
|---|---|---|---|---|---|---|---|---|---|
| | 10s | 100s | 1000s | 10s | 100s | 1000s | 10s | 100s | 1000s |
| Gurobi | 1031.39 | 87.09 | 86.52 | 682.02 | 684.12 | 684.13 | 22090.76 | 22242.58 | 22272.03 |
| PS+Gurobi | 131.87 | 125.26 | 125.26 | **684.13** | **684.13** | 684.13 | 22140.65 | 22243.12 | 22272.47 |
| DiffILO+Gurobi | **95.65** | **86.78** | **86.48** | 684.00 | 684.12 | **684.14** | **22177.82** | **22260.48** | **22272.55** |
| SCIP | 96.15 | 89.91 | 86.93 | 660.79 | 679.80 | 684.05 | 21013.73 | 22151.71 | **22272.55** |
| PS+SCIP | 125.26 | 125.26 | 125.26 | 664.50 | **684.09** | 684.13 | 21712.73 | 22248.55 | **22272.55** |
| DiffILO+SCIP | **94.16** | **87.47** | **86.57** | **674.30** | 684.06 | 684.13 | **21948.70** | **22256.08** | **22272.55** |

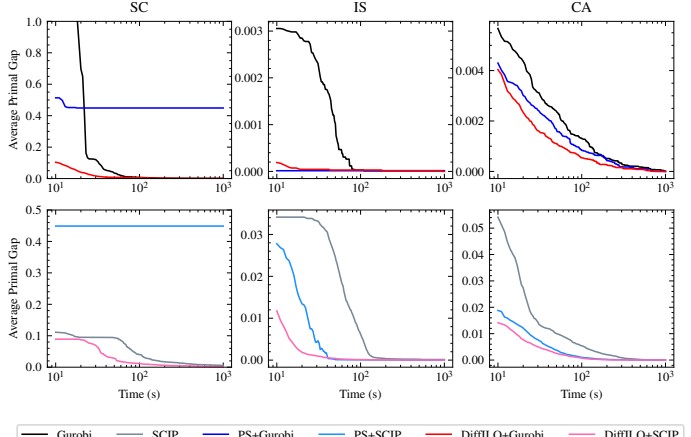

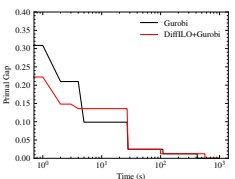

(a) `cvs16r106-72`.

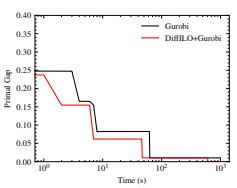

(b) `cvs16r128-89`.

Figure 5: **The relative primal gap of different approaches as the solving process proceeds.** The results are averaged across 100 test instances.

Figure 6: The relative primal gap on two cvs test instances.

**Training and Inference** We train DiffILO for 1, 200 epochs on the SC and IS datasets, and 2, 400 epochs on the CA dataset. After each epoch, we use the trained model to generate solutions for the validation instances, recording the best feasible solutions and selecting the best epoch based on their average objectives. We train the PS predictor for 2, 400 epochs on all datasets, selecting the best epoch based on validating prediction loss. To improve traditional solvers, PS adds constraints to restrict the solution space within a trust region, which is controlled by some key hyperparameters. Finding suitable hyperparameters on different datasets is challenging and labor-intensive. In contrast, as shown in Formula 9, DiffILO employs a simpler approach that does not require extensive hyperparameter tuning. In our experiments, we simply set $\Delta = 200$. More details about the training and inference processes can be found in Appendix C.2 and Appendix C.3, respectively.

**Training Time Comparison** Figure 3 shows a comparison of the training times for DiffILO and PS. The results show that supervised learning methods like PS spends much more time on collecting training labels than training the neural networks. In contrast, DiffILO bypasses the labor-intensive labeling process, achieving an average speedup of 13.2 times across the three datasets.

**Generating Feasible Solutions** We evaluate the ability of different methods to generate high-quality feasible solutions. The evaluation is performed on the 100 test instances. In Figure 4, for PS and DiffILO, we sample 30 solutions from the predicted distribution for each instance, select the best feasible solution, and report the obtained objective value. The average feasible ratio for DiffILO, computed as $(\sum_{\text{instnace}} \frac{\#\text{feasible}}{30})/(\#\text{instances})$, is 50.8%, 97.1%, and 99.4%, on SC, IS, and CA datasets, respectively. The results show that DiffILO consistently produces high-quality feasible solutions even without solver assistance on almost all instances, while PS struggles to generate feasible solutions on many instances. Gurobi has a heuristic module to find a heuristic solution before

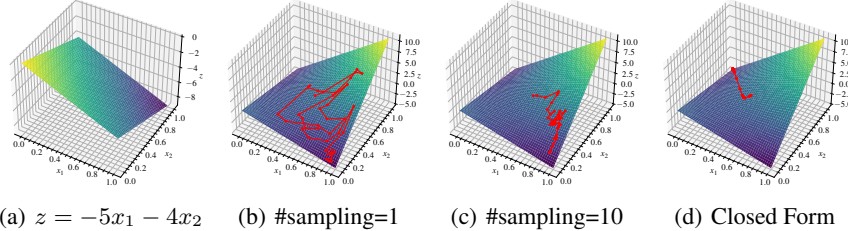

(a) $z = -5x_1 - 4x_2$     (b) #sampling=1     (c) #sampling=10     (d) Closed Form

Figure 7: **An illustrative example.** (a) shows the objective function. (b) and (c) visualize the optimization processes of DiffILO, which converge to the optimal solution. (d) visualizes the optimization process of optimizing the closed-form objective, which converges to a sub-optimal solution.

executing the exact resolution process. We further assess the solutions found by combining these methods with this heuristic module. Results show that both PS and DiffILO can improve Gurobi's heuristic solutions, but DiffILO+Gurobi significantly outperforms PS+Gurobi, with objectives very close to the best known solutions.

**Improving traditional Solvers** We then evaluate the capability of the generated solutions to accelerate the traditional solvers like Gurobi and SCIP to find better solutions in a constrained time frame. The results are in Table 1 and the solving processes are in Figure 5. The results demonstrate that DiffILO consistently outperforms PS on the SC and CA datasets. We present the solving curves on the CVS dataset in Figure 6, and report the solving time on 3 neos test instances in Table 7 in Appendix D.3. Other additional experiments, including ablation studies and additional baselines, can be found in Appendix D.

**Case Study** We present an illustrative example to demonstrate DiffILO's optimization process. We consider a simple ILP problem

$$\min_{x_1, x_2 \in \{0,1\}} \left\{ -5x_1 - 4x_2 \mid x_1 + x_2 \leq 1 \right\}. \tag{10}$$

This case is non-trivial for gradient descent, as its optimal solution $(1, 0)$ and an sub-optimal solution $(0, 1)$ have very close objectives. Figure 7 (a) displays the objective function $z = -5x_1 - 4x_2$. The transformed continuous unconstrained optimization problem is:

$$\min_{\hat{x}_1, \hat{x}_2 \in [0,1]} \left\{ F_\mu(\hat{x}_1, \hat{x}_2) \triangleq -5\hat{x}_1 - 4\hat{x}_2 + \mu \cdot \mathbb{E}_{x_1, x_2} \left[ \max(x_1 + x_2 - 1, 0) \right] \right\}, \tag{11}$$

where $x_i \sim Bernoulli(\hat{x}_i)$. The closed form of $F_\mu(\hat{x}_1, \hat{x}_2)$ is derived as

$$F_\mu(\hat{x}_1, \hat{x}_2) = -5\hat{x}_1 - 4\hat{x}_2 + \mu \cdot \hat{x}_1 \hat{x}_2. \tag{12}$$

We first set $\mu = 20$. Figures 7 (b) and (c) visualize the optimization process of DiffILO, which samples $x_1$ and $x_2$ with the reparameterization trick for optimization. The numbers of sampled solutions (#sampling) are 1 and 10 in (b) and (c), respectively. They both converge to the optimal solution $(1, 0)$. Moreover, increasing the number of samples stabilizes the optimization process. However, as shown in Figure 7 (d), when we directly optimize the smoothed function, i.e., the closed form of $F_\mu$, it converges to a sub-optimal solutions $(0, 1)$. We conduct experiments with 20 different random seeds. In 11 out of 20 runs, the optimization of the closed-form objective derives sub-optimal solutions. In contrast, DiffILO's optimization approach derives the optimal solution in all the 20 runs. However, using the closed-form penalty function will always perform penalties. We further conduct experiments to investigate the influence of $\mu$, and the results are shown in Table 9 in Appendix D.8. More discussions about this paper can be found in Appendix E.

## 5 CONCLUSION

This paper proposes **DiffILO**, a novel **Diff**erentiable **I**nteger **L**inear Programming **O**ptimization approach for learning to predict ILP solutions under an unsupervised learning paradigm. It is to our knowledge the first method that employs pure ML techniques to solve general ILPs entirely without the aid of traditional solvers. Experiments on commonly used ILP datasets demonstrate the effectiveness of DiffILO in reducing training time and producing high-quality solutions.

## REPRODUCIBILITY STATEMENT

We provide the following information for the reproducibility of our proposed DiffILO. The method is detailed in Section 3, with the proofs for theorems available in Appendix A. The implementation details, including data representation and model architecture, are provided in Appendix B The experimental details and results are in Section 4 and further elaborated in Appendix C. The code is publicly available at `https://github.com/MIRALab-USTC/L2O-DiffILO`.

## ACKNOWLEDGMENTS

The authors would like to thank all the anonymous reviewers for their insightful comments. This work was supported in part by the National Key R&D Program of China under contract 2022ZD0119801, National Nature Science Foundations of China grants U23A20388, and 62021001. This work was supported in part by Huawei as well.

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

# A    PROOFS

**Theorem 1.** *The problem (P2) is feasible (and solvable, i.e., it admits at least one optimal solution) if and only if (P1) is feasible (and solvable).*

*Proof.* First, we show the equivalence of the feasibility. It is obvious that any feasible solution $\boldsymbol{x}$ to (P1) is also a feasible solution to (P2). Conversely, let us consider $\hat{\boldsymbol{x}}^*$ as a solution to (P2). It follows that $\mathbb{E}_{\boldsymbol{x} \sim p(\cdot | \hat{\boldsymbol{x}}^*)}[\max(\boldsymbol{A}\boldsymbol{x} - \boldsymbol{b}, \boldsymbol{0})] = \boldsymbol{0}$. Notably, $\operatorname{supp} p(\cdot | \hat{\boldsymbol{x}}^*)$ is not empty, and for $\forall \boldsymbol{x} \in \operatorname{supp} p(\cdot | \hat{\boldsymbol{x}}^*)$, we have $\max(\boldsymbol{A}\boldsymbol{x} - \boldsymbol{b}, \boldsymbol{0}) = \boldsymbol{0}$, and thus $\boldsymbol{A}\boldsymbol{x} \leq \boldsymbol{b}$. Consequently, $\boldsymbol{x}$ is a feasible solution to (P1).

Next, we show the equivalence of solvability. Notice that the domain of (P1) is $\{0, 1\}^n$, which is a finite set. Therefore, (P1) is solvable if and only if it is feasible. Similarly, as the objective of (P2) is a continuous function over the compact set $[0, 1]^n$, (P2) is solvable if and only if it is feasible. Based on these observations, we conclude that (P2) is solvable if and only if (P1) is solvable. □

**Theorem 2.** *Let $\mathcal{I}_{\boldsymbol{c}} \triangleq \{i \in [n] : c_i \neq 0\}$. Then the following statements hold:*

1. *Suppose $\boldsymbol{x}^* \in \{0, 1\}^n$ is an optimal solution to (P1). Then $\boldsymbol{x}^*$ is also an optimal solution to (P2). If a vector $\hat{\boldsymbol{x}}^* \in [0, 1]^n$ is a feasible solution to (P2) and satisfies $\hat{x}_i^* = x_i^*$ for all $i \in \mathcal{I}_{\boldsymbol{c}}$, then $\hat{\boldsymbol{x}}^*$ is an optimal solution to (P2).*

2. *Suppose $\hat{\boldsymbol{x}}^* \in [0, 1]^n$ is an optimal solution to (P2). Then we have $\hat{x}_i^* \in \{0, 1\}$ for all $i \in \mathcal{I}_{\boldsymbol{c}}$. Let $\mathcal{I}_{\hat{\boldsymbol{x}}^*} = \{i \in [n] : \hat{x}_i^* \in \{0, 1\}\}$. If a vector $\boldsymbol{x}^* \in \{0, 1\}^n$ satisfies $x_i^* = \hat{x}_i^*$ for all $i \in \mathcal{I}_{\hat{\boldsymbol{x}}^*}$, then $\boldsymbol{x}^*$ is an optimal solution to (P1).*

*Proof.* 1. Let $\boldsymbol{x}^* \in \{0, 1\}^n$ be an optimal solution to (P1). Obviously it is also a feasible solution to (P2). Assume $\hat{\boldsymbol{x}}^* \in [0, 1]^n$ is a feasible solution to (P2) and satisfies $\hat{x}_i^* = x_i^*$ for all $i \in \mathcal{I}_{\boldsymbol{c}}$. It follows that $\boldsymbol{c}^\top \hat{\boldsymbol{x}}^* = \boldsymbol{c}^\top \boldsymbol{x}^*$. Let $\hat{\boldsymbol{x}} \in [0, 1]^n$ be any feasible solution to (P2). For $\forall \boldsymbol{x} \in \operatorname{supp} p(\cdot | \hat{\boldsymbol{x}})$, we have $\max(\boldsymbol{A}\boldsymbol{x} - \boldsymbol{b}, \boldsymbol{0}) = \boldsymbol{0}$, and thus $\boldsymbol{A}\boldsymbol{x} \leq \boldsymbol{b}$, which indicates that $\boldsymbol{x}$ is a feasible solution to (P1) and thus $\boldsymbol{c}^\top \boldsymbol{x}^* \leq \boldsymbol{c}^\top \boldsymbol{x}$. Then we have $\boldsymbol{c}^\top \boldsymbol{x}^* \leq \boldsymbol{c}^\top \mathbb{E}_{\boldsymbol{x} \sim p(\cdot | \hat{\boldsymbol{x}})}[\boldsymbol{x}] = \boldsymbol{c}^\top \hat{\boldsymbol{x}}$. It follows that $\boldsymbol{c}^\top \hat{\boldsymbol{x}}^* = \boldsymbol{c}^\top \boldsymbol{x}^* \leq \boldsymbol{c}^\top \hat{\boldsymbol{x}}$, indicating that $\hat{\boldsymbol{x}}^*$ is an optimal solution to (P2). This conclusion, combined with the feasibility of $\boldsymbol{x}^*$, gives that $\boldsymbol{x}^*$ is also an optimal solution to (P2).

2. Let $\hat{\boldsymbol{x}}^* \in [0, 1]^n$ be an optimal solution to (P2). Consider if $\hat{x}_i^* \in (0, 1)$ for some $i \in \mathcal{I}_{\boldsymbol{c}}$. Without loss of generality, we assume $c_i > 0$. Define $\hat{\boldsymbol{x}}'$ such that $\hat{x}_i' = 0$ and $\hat{x}_j' = \hat{x}_j$ for $j \neq i$. Then $\operatorname{supp} p(\cdot | \hat{\boldsymbol{x}}') \subset \operatorname{supp} p(\cdot | \hat{\boldsymbol{x}}^*)$, and so $\hat{\boldsymbol{x}}'$ remains a feasible solution to (P2). However, $\boldsymbol{c}^\top \hat{\boldsymbol{x}}' < \boldsymbol{c}^\top \hat{\boldsymbol{x}}^*$, contradicting the optimality of $\hat{\boldsymbol{x}}^*$. Therefore $\hat{x}_i^* \in \{0, 1\}$ for all $i \in \mathcal{I}_{\boldsymbol{c}}$.

Let $\boldsymbol{x}^* \in \{0, 1\}^n$ satisfies $x_i^* = \hat{x}_i^*$ for all $i \in \mathcal{I}_{\hat{\boldsymbol{x}}^*}$. Then $\boldsymbol{x}^* \in \operatorname{supp} p(\cdot | \hat{\boldsymbol{x}}^*)$ and thus it is a feasible solution to (P1). We also have $\boldsymbol{c}^\top \boldsymbol{x}^* = \boldsymbol{c}^\top \hat{\boldsymbol{x}}^*$. Let $\forall \boldsymbol{x} \in \{0, 1\}^n$ be a feasible solution to (P1). Then it is also a feasible solution to (P2), indicating that $\boldsymbol{c}^\top \hat{\boldsymbol{x}}^* \leq \boldsymbol{c}^\top \boldsymbol{x}$. In follows that $\boldsymbol{c}^\top \boldsymbol{x}^* = \boldsymbol{c}^\top \hat{\boldsymbol{x}}^* \leq \boldsymbol{c}^\top \boldsymbol{x}$, indicating that $\boldsymbol{x}^*$ is an optimal solution to (P1). □

**Theorem 3.** *There exists a positive scalar $\mu^* > 0$ such that for any $\mu > \mu^*$, any optimal solution to (P3) is also an optimal solution to (P2).*

*Proof.* We define $\phi(\boldsymbol{x}) \triangleq \sum_j \phi_j(\boldsymbol{x})$ and $\hat{\phi}(\hat{\boldsymbol{x}}) \triangleq \sum_j \hat{\phi}_j(\hat{\boldsymbol{x}})$, which leads to $\hat{\phi}(\hat{\boldsymbol{x}}) = \mathbb{E}_{\boldsymbol{x} \sim p(\cdot | \hat{\boldsymbol{x}})}[\phi(\boldsymbol{x})]$. The domain of $\phi(\cdot)$ is $\{0, 1\}^n$, a finite set. Therefore, we can find

$$\rho \triangleq \min_{\boldsymbol{x} \in \operatorname{supp} \phi(\cdot)} \phi(\boldsymbol{x}). \tag{13}$$

It follows that $\phi(\boldsymbol{x}) \geq \rho$ for $\forall \boldsymbol{x} \in \operatorname{supp} \phi(\cdot)$. We set $\mu^* = \frac{2\sqrt{n} \|\boldsymbol{c}\|_2}{\rho}$, and assume $\mu > \mu^*$. Let $\hat{\boldsymbol{x}}^* \in [0, 1]^n$ be an optimal solution to (P3) and $\hat{\boldsymbol{x}}$ be any feasible solution to (P3). It suffices to show that $\hat{\phi}(\hat{\boldsymbol{x}}^*) = \sum_j \hat{\phi}_j(\hat{\boldsymbol{x}}^*) = 0$ and that $\boldsymbol{c}^\top \hat{\boldsymbol{x}}^* \leq \boldsymbol{c}^\top \hat{\boldsymbol{x}}$.

We denote $\mathcal{X}^* \triangleq \arg\min_{\boldsymbol{x} \in \operatorname{supp} p(\cdot | \hat{\boldsymbol{x}}^*)} \{\boldsymbol{c}^\top \boldsymbol{x} + \mu \cdot \phi(\boldsymbol{x})\}$. The optimality of $\hat{\boldsymbol{x}}^*$ indicates that $\operatorname{supp} p(\cdot | \hat{\boldsymbol{x}}^*) = \mathcal{X}^*$, and that

$$\boldsymbol{c}^\top \hat{\boldsymbol{x}}^* + \mu \cdot \hat{\phi}(\hat{\boldsymbol{x}}^*) \leq \boldsymbol{c}^\top \hat{\boldsymbol{x}} + \mu \cdot \hat{\phi}(\hat{\boldsymbol{x}}). \tag{14}$$

The feasibility of $\hat{x}$ indicates that $\hat{\phi}(\hat{x}) = 0$, and thus

$$c^\top \hat{x}^* + \mu \cdot \hat{\phi}(\hat{x}^*) \leq c^\top \hat{x}. \tag{15}$$

Therefore, for $\forall x^* \in \mathcal{X}^* = \text{supp}\, p(\cdot|\hat{x}^*)$, we have

$$\mu \cdot \phi(x^*) \leq c^\top (\hat{x} - x^*) \leq \|c\|_2 (\|\hat{x}\|_2 + \|x^*\|_2) \leq 2\sqrt{n}\|c\|_2. \tag{16}$$

Consider if $\hat{\phi}(\hat{x}^*) > 0$, which implies that $\text{supp}\, p(\cdot|\hat{x}^*) \cap \text{supp}\, \phi(\cdot) \neq \emptyset$. Let $x^* \in \text{supp}\, p(\cdot|\hat{x}^*) \cap \text{supp}\, \phi(\cdot)$. Then we have

$$\mu^* < \mu \leq \frac{c^\top (\hat{x} - x^*)}{\phi(x^*)} \leq \frac{\|c\|_2 (\|\hat{x}\|_2 + \|x^*\|_2)}{\rho} \leq \frac{2\sqrt{n}\|c\|_2}{\rho} = \mu^*, \tag{17}$$

leading to a contradiction. Therefore, we have $\hat{\phi}(\hat{x}^*) = 0$. Plugging it into Equation 14 completes the proof. $\quad\square$

**Lemma 1.** *(Restated from (Maddison et al., 2016))* *Let $\hat{x} \in (0,1)$ and $\epsilon$ be a random variable sampled from $\mathcal{U}(0,1)$. We define $\xi(\hat{x}; \epsilon) \triangleq \sigma(\tau(\hat{x}) + \tau(\epsilon))$. It follows that $P(\xi(\hat{x}; \epsilon) > 0.5) = \hat{x}$.*

*Proof.* We have

$$
\begin{aligned}
&P\left(\xi(\hat{x}; \epsilon) > 0.5\right) \\
=&P\left(\log(\frac{\hat{x}}{1-\hat{x}}) + \log(\frac{\epsilon}{1-\epsilon}) > 0\right) \\
=&P\left(\frac{\hat{x}}{1-\hat{x}} \cdot \frac{\epsilon}{1-\epsilon} > 1\right) \\
=&P\left(\epsilon > 1 - \hat{x}\right) = \hat{x}.
\end{aligned}
\tag{18}
$$

$\square$

**Theorem 4.** *Let $\hat{x} = (\hat{x}_1, \cdots, \hat{x}_n) \in (0,1)^n$, and $\epsilon = (\epsilon_1, \cdots, \epsilon_n)^\top$ be a random vector, where each $\epsilon_i$ is independently and identically distributed (i.i.d.) as $\epsilon_i \sim \mathcal{U}(0,1)$, writing $\epsilon \sim p_\epsilon(\cdot)$. Let $\xi(\hat{x}; \epsilon) \triangleq (\xi_1, \cdots, \xi_n)^\top$, where $\xi_i = \xi(\hat{x}_i; \epsilon_i)$ is defined as in Lemma 1. Let $\psi(\hat{x}; \epsilon) \triangleq (\psi_1, \cdots, \psi_n)^\top$, where $\psi_i = [\xi_i]$ is the binary rounded value of $\xi_i$. It follows that:*

$$\hat{\phi}_j(\hat{x}) = \mathbb{E}_{x \sim p(\cdot|\hat{x})}[\phi_j(x)] = \mathbb{E}_{\epsilon \sim p_\epsilon(\cdot)}[\phi_j(\psi(\hat{x}; \epsilon))]. \tag{1}$$

*Proof.* By Lemma 1, we have

$$P(\psi_i = 1) = P(\xi_i > 0.5) = \hat{x}_i, \tag{19}$$

which implies that $p(x|\hat{x}) = p(\psi(\hat{x}; \epsilon) = x|\hat{x})$. Therefore,

$$
\begin{aligned}
\hat{\phi}_j(\hat{x}) &= \mathbb{E}_{x \sim p(\cdot|\hat{x})}[\phi_j(x)] \\
&= \sum_{x \in \{0,1\}^n} \phi_j(x) p(x|\hat{x}) \\
&= \sum_{x \in \{0,1\}^n} \phi_j(x) p(\psi(\hat{x}; \epsilon) = x|\hat{x}) \\
&= \sum_{x \in \{0,1\}^n} \phi_j(x) \int_{\epsilon : x = \psi(\hat{x}; \epsilon)} p(x|\hat{x}, \epsilon) p_\epsilon(\epsilon) d\epsilon \\
&= \int_\epsilon \left(\sum_{x = \psi(\hat{x}; \epsilon)} \phi_j(x)\right) p_\epsilon(\epsilon) d\epsilon \\
&= \mathbb{E}_{\epsilon \sim p_\epsilon(\cdot)}[\phi_j(\psi(\hat{x}; g))].
\end{aligned}
\tag{20}
$$

$\square$

**Theorem 5.** *The merit function $\mathcal{F}_\mu(\hat{\boldsymbol{x}})$ defined in (P4) is differentiable almost everywhere (a.e.) in $(0, 1)^n$. At the differentiable points, the gradient is given by:*

$$\nabla_{\hat{\boldsymbol{x}}} \mathcal{F}_\mu(\hat{\boldsymbol{x}}) = \boldsymbol{c} + \mu \sum_{j=1}^m \int_{\boldsymbol{\epsilon}: \boldsymbol{a}_j^\top \boldsymbol{\psi}(\hat{\boldsymbol{x}}; \boldsymbol{\epsilon}) - b_j > 0} \boldsymbol{a}_j \odot \left( \frac{\partial}{\partial \hat{\boldsymbol{x}}} \odot \boldsymbol{\xi}(\hat{\boldsymbol{x}}; \boldsymbol{\epsilon}) \right) p_{\boldsymbol{\epsilon}}(\boldsymbol{\epsilon}) \mathrm{d}\boldsymbol{\epsilon}, \tag{5}$$

*where $\odot$ denotes the element-wise product.*

*Proof.* We have

$$\nabla_{\hat{\boldsymbol{x}}} \mathcal{F}_\mu(\hat{\boldsymbol{x}}) = \nabla_{\hat{\boldsymbol{x}}} \left( \boldsymbol{c}^\top \hat{\boldsymbol{x}} + \mu \sum_{j=1}^m \hat{\varphi}_j(\hat{\boldsymbol{x}}) \right)$$

$$= \boldsymbol{c} + \mu \sum_{j=1}^m \nabla_{\hat{\boldsymbol{x}}} \hat{\varphi}_j(\hat{\boldsymbol{x}}) = \boldsymbol{c} + \mu \sum_{j=1}^m \nabla_{\hat{\boldsymbol{x}}} \mathbb{E}_{\boldsymbol{\epsilon} \sim p_{\boldsymbol{\epsilon}}(\cdot)} \left[ \varphi_j(\hat{\boldsymbol{x}}; \boldsymbol{\epsilon}) \right]$$

$$= \boldsymbol{c} + \mu \sum_{j=1}^m \mathbb{E}_{\boldsymbol{\epsilon} \sim p_{\boldsymbol{\epsilon}}(\cdot)} \left[ \frac{\partial}{\partial \hat{\boldsymbol{x}}} \left( \left( \boldsymbol{a}_j^\top \boldsymbol{\xi}(\hat{\boldsymbol{x}}; \boldsymbol{\epsilon}) - b_j \right) \mathbb{I} \left( \boldsymbol{a}_j^\top \boldsymbol{\psi}(\hat{\boldsymbol{x}}; \boldsymbol{\epsilon}) - b_j > 0 \right) \right) \right] \tag{21}$$

$$= \boldsymbol{c} + \mu \sum_{j=1}^m \int_{\boldsymbol{\epsilon}: \boldsymbol{a}_j^\top \boldsymbol{\psi}(\hat{\boldsymbol{x}}; \boldsymbol{\epsilon}) - b_j > 0} \frac{\partial}{\partial \hat{\boldsymbol{x}}} \left( \boldsymbol{a}_j^\top \boldsymbol{\xi}(\hat{\boldsymbol{x}}; \boldsymbol{\epsilon}) - b_j \right) p_{\boldsymbol{\epsilon}}(\boldsymbol{\epsilon}) \mathrm{d}\boldsymbol{\epsilon}$$

$$= \boldsymbol{c} + \mu \sum_{j=1}^m \int_{\boldsymbol{\epsilon}: \boldsymbol{a}_j^\top \boldsymbol{\psi}(\hat{\boldsymbol{x}}; \boldsymbol{\epsilon}) - b_j > 0} \boldsymbol{a}_j \odot \left( \frac{\partial}{\partial \hat{\boldsymbol{x}}} \odot \boldsymbol{\xi}(\hat{\boldsymbol{x}}; \boldsymbol{\epsilon}) \right) p_{\boldsymbol{\epsilon}}(\boldsymbol{\epsilon}) \mathrm{d}\boldsymbol{\epsilon}.$$

$\square$

## B  IMPLEMENTATION DETAILS

### B.1  DATA REPRESENTATION

Following previous works (Gasse et al., 2019; Han et al., 2023; Geng et al., 2023; Wang et al., 2023b), we represent each ILP problem as a weighted bipartite graph $\mathcal{G} = (\mathcal{V} \cup \mathcal{W}, \mathcal{E})$, where $\mathcal{V}$ and $\mathcal{W}$ denote the sets of variables and constraints, respectively. The graph is equipped with a tuple of feature matrices $(\mathbf{V}, \mathbf{W}, \mathbf{E})$, and the description of these features can be found in Table 2.

Table 2: Description of variable, constraint, and edge features in our bipartite graph representation.

| Tensor | Feature | Description |
|---|---|---|
|  | Objective | Normalized objective coefficient. |
|  | Variable coefficient | Average variable coefficient in all constraints. |
| **V** | Variable degree | Degree of the variable node in the bipartite graph representation. |
|  | Maximum variable coefficient | Maximum variable coefficient in all constraints. |
|  | Minimum variable coefficient | Minimum variable coefficient in all constraints. |
|  | Constraint coefficient | Average of all coefficients in the constraint. |
| **W** | Constraint degree | Degree of constraint nodes. |
|  | Bias | Normalized right-hand-side of the constraint. |
| **E** | Coefficient | Constraint coefficient. |

### B.2  MODEL ARCHITECTURE

We employ a graph neural network (GNN), parameterized by $\boldsymbol{\theta}$, as the predictor. Specifically, given a bipartite graph $\mathcal{G} = (\mathcal{V} \cup \mathcal{W}, \mathcal{E})$ equipped with the feature metrices $(\mathbf{V}, \mathbf{W}, \mathbf{E})$, we use MLPs as embedding layers to obtain the initial embeddings :

$$\boldsymbol{h}_{v_i}^{(0)} = \mathrm{MLP}_{\boldsymbol{\theta}}(\boldsymbol{v}_i), \quad \boldsymbol{h}_{w_j}^{(0)} = \mathrm{MLP}_{\boldsymbol{\theta}}(\boldsymbol{w}_j), \quad \boldsymbol{h}_{e_{ij}} = \mathrm{MLP}_{\boldsymbol{\theta}}(\boldsymbol{e}_{ij}). \tag{22}$$

After that, we perform $K$ graph convolution layers, with each layer in the form of two interleaved half-convolutions (Gasse et al., 2019), defined as follows:

$$\boldsymbol{h}_{w_i}^{(k+1)} \leftarrow \mathrm{MLP}_{\boldsymbol{\theta}} \left( \boldsymbol{h}_{w_i}^{(k)}, \sum_{j:e_{ij} \in \mathcal{E}} \mathrm{MLP}_{\boldsymbol{\phi}} \left( \boldsymbol{h}_{w_i}^{(k)}, \boldsymbol{h}_{e_{ij}}, \boldsymbol{h}_{v_j}^{(k)} \right) \right),$$

$$\boldsymbol{h}_{v_j}^{(k+1)} \leftarrow \mathrm{MLP}_{\boldsymbol{\phi}} \left( \boldsymbol{h}_{v_j}^{(k)}, \sum_{i:e_{ij} \in \mathcal{E}} \mathrm{MLP}_{\boldsymbol{\phi}} \left( \boldsymbol{h}_{w_i}^{(k+1)}, \boldsymbol{h}_{e_{ij}}, \boldsymbol{h}_{v_j}^{(k)} \right) \right). \tag{23}$$

Each convolution layer is followed by two GraphNorm layers, one for variables and the other for constraints. We employ a concatenation Jumping Knowledge layer to aggregate information from all $K$ layers and obtain the final node representations:

$$\boldsymbol{h}_{v_i} = \mathrm{MLP}_{\boldsymbol{\theta}} \left( \underset{k=0,\cdots,K}{\mathrm{CONCAT}} \left( \boldsymbol{h}_{v_i}^{(k)} \right) \right), \quad \boldsymbol{h}_{w_j} = \mathrm{MLP}_{\boldsymbol{\theta}} \left( \underset{k=0,\cdots,K}{\mathrm{CONCAT}} \left( \boldsymbol{h}_{w_j}^{(k)} \right) \right). \tag{24}$$

Subsequently, we use another MLP to output the predicted logits for each variable:

$$\boldsymbol{z}_{v_i} = \mathrm{MLP}_{\boldsymbol{\theta}} \left( \boldsymbol{h}_{v_i} \right). \tag{25}$$

The logits are then used for resampling module, followed by a sigmoid function to output the generated solutions.

## C   EXPERIMENTAL DETAILS

### C.1   DETAILS OF THE BENCHMARKS

We use three commonly used ILP benchmarks in our experiments. The data instances are generated using the code from `https://github.com/ds4dm/learn2branch`. We list the benchmark information in Table 3, including the generation algorithms, average numbers of constraints and average numbers of variables.

Table 3: Statistics of the benchmarks.

| Dataset | Generation | Number of Constraints | Number of Variables |
|---|---|---|---|
| SC | (Balas & Ho, 1980) | 3000 | 2000 |
| IS | (Bergman et al., 2015) | 5943 | 1500 |
| CA | (Leyton-Brown et al., 2000) | 576 | 1500 |

### C.2   TRAINING DETAILS

As mentioned in Section 3.3 in the main text, here we introduce three useful techniqques in our training process.

**Normalization**   In practice, we conduct a normalization and modify the loss function on each instance $\mathcal{G}_i$ as

$$
\mathcal{L}(\boldsymbol{\theta}; \mathcal{G}_i) \triangleq \begin{cases} \dfrac{\boldsymbol{c}^\top}{\|\boldsymbol{c}\|_2} \boldsymbol{f_\theta}(\mathcal{G}_i) + \mu \sum_{j=1}^{m_i} \sum_{k=1}^{K} \dfrac{\varphi_{i,j}(\boldsymbol{f_\theta}(\mathcal{G}_i); \boldsymbol{\epsilon}_i^{(k)})}{M_i \|\boldsymbol{a}_{i,j}\|_2}, & M_i > 0, \\[4mm] \dfrac{\boldsymbol{c}^\top}{\|\boldsymbol{c}\|_2} \boldsymbol{f_\theta}(\mathcal{G}_i), & M_i = 0, \end{cases}
\tag{26}
$$

where $M_i$ is the number of constraints violated.

**Learning Rate Annealing**   To facilitate a continuing model optimization and alleviate local optimum, we adopt a cosine annealing scheduler for the learning rate (Loshchilov & Hutter, 2016), with a period denoted as lr_T. The training curves in Figure 10 in Appendix D.4 demonstrate the influece of the cosine annealing of learning rate on the training progress.

**Adaptive Penalty Coefficient**   The penalty coefficient $\mu$ is an important hyperparameter in DiffILO, which influences the convergence of the training process. It is not set on per-instance level, but setting it as a single value is enough. Our probability modeling approach can somehow reduce the influence of $\mu$. Specifically, the penalty term in our method is defined as $\sum_j \mathbb{E}_{x \sim p(\cdot|\hat{x})}[\max(a_j^\top x - b_j, 0)]$. Notice that the penalty is only activated when when the constraint is violated. Thus, even if the penalty parameter is set relatively large, the penalty term is less likely to dominate the loss function significantly if the constraints are not violated.

To reduce the need for manual parameter adjustment for $\mu$, we use a dynamic and adaptive $\mu$, which is inspired by the adaptive temperature in soft actor-critic algorithm (Haarnoja et al., 2018). Specifically, after each epoch, we update the coefficient $\mu$ according to the updating rule

$$
\mu_{k+1} = \mu_k + \text{mu\_step} * (\text{cons} - \text{cons\_targ}),
\tag{27}
$$

where cons denotes the average constraint violation in this epoch, and cons_targ is the target value of the average constraint violation. Empirically the hyperparameter mu_targ is set as no more than 1 (according to the range of coefficients), as this indicates that there exist solutions with no constraint violation in a probabilistic sense. This dynamic way for tuning $\mu$ can effectively improve the algorithm robustness against the choice of $\mu$. We present the training curves with different values the parameter $\mu$ and analyze the influence of the adaptive strategy for $\mu$. The results are in Appendix D.5.

Some important hyperparameters of the model training are provided in Table 4.

Table 4: Hyperparameters in our experiments.

| Hyperparameter | SC | IS | CA | Description |
|---|---|---|---|---|
| embed_size | 32 | 32 | 32 | The embedding size of the GNN predictor. |
| depth | 3 | 10 | 10 | The depth of the GNN predictor. |
| batch_size | 5 | 5 | 5 | Number of ILP problems in each training batch. |
| num_samples | 15 | 15 | 15 | Number of sampled solutions for reparameterization. |
| num_epochs | 1,200 | 1,200 | 2,400 | Number of max running epochs. |
| optimizer | Adam | Adam | Adam | Optimizer for training. |
| learning_rate | 1e-4 | 8e-5 | 8e-5 | Leaning rate for training. |
| lr_T | 200 | 200 | 200 | The period for learning rate cosine annealing. |
| mu_init | 5.0 | 100.0 | 15 | The initial value of $\mu$. |
| mu_step | 0.01 | 1.0 | 0.001 | Step size for optimizing $\mu$. |
| cons_targ | 1.0 | 0.1 | 0.1 | Target value of average constraint violation. |

## C.3 INFERENCE DETAILS

To incorporate the heuristic solutions into traditional solvers like Gurobi and SCIP, we define a trust region by limiting the number of variables—which are different from the predicted ones—to be fewer than 200. In practice, we add the following constraint to the original problem

$$\sum_{\hat{x}_i=0} x_i + \sum_{\hat{x}_i=1} (1 - x_i) < \Delta, \tag{28}$$

where $\hat{x}$ denotes the provided heuristic solution, and $\Delta$ is a hyperparameter. In our experiments, we simply set $\Delta = 200$.

We also supply our best-found solution as an initial solution to the solver. For Gurobi, this is implemented as:

```
for i, v in enumerate(m.getVars()):
    v.Start = best_x[i]
```

and for SCIP it is implemented as:

```
sol = m.createPartialSol()
for i, v in enumerate(m.getVars()):
    m.setSolVal(sol, v, best_x[i])
    m.addSol(sol)
```

Only one best solution is provided to the solver. Specifically, for each instance, we sample 1,000 solutions from the predicted distribution. We select the best feasible solution, if any, based on the objective value. If no feasible solution found, we select the one with the minimal merit function.

During inference, SCIP 8.1.0 (Achterberg, 2009), Gurobi 10.0.1 (Gurobi Optimization, 2021) are used for solving instances. Following Han et al. (2023), we configure the solvers towards the "heuristic-first" mode—the "MIPFocus" parameter for Gurobi and the "AGGRESSIVE" parameter in SCIP—so that they will focus on finding better primal solutions. Specifically, we set 'm.Params.MIPFocus = 1' for Gurobi and 'm.setHeuristics(SCIP_PARAMSETTING.AGGRESSIVE)' for SCIP, respectively. The time limit for running each experiment is set to $1,000$ seconds.

To improve traditional solvers, PS (Han et al., 2023) adds constraints to restrict the solution space within a trust region. The trust region search algorithm is controlled by three key hyperparameters, $k_0, k_1$, and $\Delta$. Searching for suitable hyperparameters for PS on different datasets is challenging and labor-intensive. For IS and CA, we use the default hyperparameters specified in the original paper. For SC, which is not included in the original paper, we conduct the hyperparameter search as follows: we first fix $k_0$ and $k_1$ to 100 and experiment with $\Delta$ values in $\{5, 10, 15, 20\}$. We then experiment with $k_0$ and $k_1$ values in $\{100, 200, 300\}$ with fixed $\Delta$.

# D  ADDITIONAL RESULTS

## D.1  SHIFTED GEOMETRIC MEAN OF RELATIVE GAPS

To better understand the results, we also report the shifted geometric mean (SGM) of relative gaps of the instances in Table 5, which is a usually used metric to measure the model performance in the ILP community. Specifically, suppose the dataset contains $N$ instances, the $i$th instance's best know objective is $\text{BKS}_i$, and a method achieves an objective value $\text{OBJ}_i$. Its relative gap is defined as

$$\text{gap}_i = \frac{|\text{OBJ}_i - \text{BKS}_i|}{|\text{BKS}_i|}. \tag{29}$$

The shifted geometric mean across all instances is defined as

$$\text{SGM} = \exp\left(\frac{1}{N}\sum_i \log(\text{gap}_i + 1.0)\right) - 1.0. \tag{30}$$

Table 5: **Shifted geometric mean (SGM) of relative gaps of different methods on all datasets.** Lower SGM indicates better performance. We mark the best results in bold and underline the second-best results.

| | SC | | | IS | | | CA | | |
|---|---|---|---|---|---|---|---|---|---|
| | 10s | 100s | 1000s | 10s | 100s | 1000s | 10s | 100s | 1000s |
| Gurobi | 10.6222 | 0.0070 | 0.0008 | 0.0031 | 2.97E-05 | 1.50E-05 | 0.0059 | 0.0013 | 3.48E-06 |
| PS+Gurobi | 0.5085 | 0.449 | 0.449 | **1.50E-05** | **1.50E-05** | 1.50E-05 | 0.0044 | 0.0009 | **3.61E-10** |
| DiffILO+Gurobi | **0.1046** | **0.0037** | **0.0005** | 0.0002 | 2.96E-05 | **0** | 0.0042 | 0.0005 | 4.10E-10 |
| SCIP | 0.11 | 0.0373 | 0.0052 | 0.0341 | 0.0063 | 0.0001 | 0.0563 | 0.0054 | 1.0067 |
| PS+SCIP | 0.449 | 0.449 | 0.449 | 0.0286 | **7.36E-05** | 1.50E-05 | 0.0249 | 0.0011 | **1.00E-07** |
| DiffILO+SCIP | **0.0873** | **0.0112** | **0.0015** | **0.0143** | 1.00E-04 | **1.50E-05** | **0.0145** | **0.0007** | **1.00E-07** |

## D.2  RESULTS OF ADDITIONAL BASELINES

In the main text, our main baseline is the PS method, which is a representative supervised learning approach. In this section, we include three additional baselines as follows, and the experimental results on the SC dataset are shown in Table 6 and Figure 8.

**CL-LNS**  (Huang et al., 2023) is a large neighborhood search (LNS) approach, a different framework from branch and bound (BnB). We compare with CL-LNS to demonstrate the effectiveness of DiffILO compared with LNS-based methods.

**ConPaS**  (Huang et al., 2024) adopts contrastive learning for enhancing the PS framework. As they have not provided publicly released code, we implement the approach based on the paper's details.

**DDIM**  (Zeng et al., 2024) adopts diffusion models to learn to generate feasible solutions. We used the authors' released code.

**Naive Relaxation**  We include a naive relaxation baseline for the ablation study to demonstrate the effectiveness of our proposed relaxation approach. Specifically, in this baseline, we directly optimize the penalty term $\sum_j \max(a_j^\top \hat{x} - b_j, 0)$ instead of our proposed probabilistic one $\sum_j \mathbb{E}_{x \sim p(\cdot|\hat{x})}[\max(a_j^\top x - b_j, 0)]$. The parameter $\mu$ is further tuned to achieve good convergence. Different from our approach, the naive baseline views the problem as a simple continuous one without considering the discrete nature of the original problem.

**REINFORCE rather than reparameterization**   As noted in Remark 5, an alternative approach for handling non-differentiable computation graphs involving sampling is the REINFORCE method. To demonstrate the effectiveness of using reparameterization, we implement a REINFORCE method as a baseline. It computes gradients as

$$\nabla_{\hat{x}}\mathbb{E}_{x\sim p(\cdot|\hat{x})}[C(x)] = \mathbb{E}_{x\sim p(\cdot|\hat{x})}[C(x)\nabla_{\hat{x}}\log p(\cdot|\hat{x})], \tag{31}$$

where $C(x)$ denotes the merit function as defined in (P3). We find that all models collapse towards minimal objectives but significant constraint violations, even if we set a very large $\mu$. This tendency underscores the well-known training challenges associated with RL models. Specifically, the REINFORCE method relies on random exploration without gradient guidance. When a solution is reached, the model receives only a reward signal but is unaware of the inherent components of the reward or the gradient information at the current point. In the vast search space, the absence of gradient-directed exploration can lead models to converge to trivial yet infeasible solutions. The results further demonstrate the necessity of re-parameterization trick for this task.

Table 6: **Results of additional baselines.** We report the objective values achieved at 10s, 100s, and 1000s, respectively. The results show that DiffILO still outperforms these baselines. Another baseline, REINFORCE, has failed to derive meaningful results and thus is not reported.

|  | SC (Min, BKS: 86.45) | | |
|  | 10s | 100s | 1000s |
| --- | --- | --- | --- |
| CL-LNS | 203.27 | 91.96 | 86.77 |
| Naive Relaxation | 132.1 | 94.71 | 87.05 |
| DiffILO | **95.65** | **86.78** | **86.48** |

We test the abilities of DiffILO, ConPaS, and DDIM to generate feaisble solutions. The results are shown in Figure 8. The results show that ConPaS still fails to generate feasible solutions across most instances. DDIM demonstrates strong feasibility rates and successfully generates feasible solutions for all instances. However, when considering solution quality, DiffILO still outperformed DDIM in terms of objective values. This highlights the strength of DiffILO in producing higher-quality solutions.

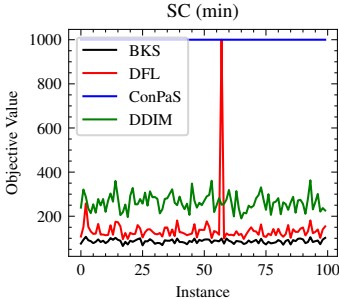

Figure 8: Results of DiffILO, ConPaS (Huang et al., 2024), and DDIM (Zeng et al., 2024) on generating feasible solutions on SC.

### D.3   RESULTS ON MIPLIB DATASETS

To demonstrate the effectiveness of DiffILO on a more complex and realistic dataset, we conduct additional experiments on the MIPLIB 2017 benchmark (Gleixner et al., 2021), which is well known as a collection of challenging real-world MILP instances. Notice that MIPLIB contains instances across many different scenarios, and many of the large-scale problems do not have isomorphic counterparts, learning directly on the full MIPLIB is extremely challenging. Following Wang et al. (2023a), we first construct a subset of MIPLIB, called "MIPLIB-CVS", to validate the effectiveness of DiffILO. Specifically, it contains five capacitated vertex separator (CVS) problem instances from MIPLIB 2017. They are `cvs08r139-94`, `cvs16r70-62`, `cvs16r89-60`, `cvs16r106-72`,

and `cvs16r128-89`. We use the first three instances for training DiffILO, and then test the model on the last two instances. The total training process takes about only 9 minutes. If we use supervised learning, it will take more than three hours to solve the training instances for providing labels. The solving progresses are shown in Figure 6, which overall demonstrate that DiffILO can accelerate the solving process even on complex realistic datasets. Notably, on the `cvs16r128-89`, DiffILO achieves the optimal solution (which is $-97.0$) in $1,000$ seconds, which even surpasses the result obtained by running Gurobi for $3,600$ seconds (which is $-96.0$).

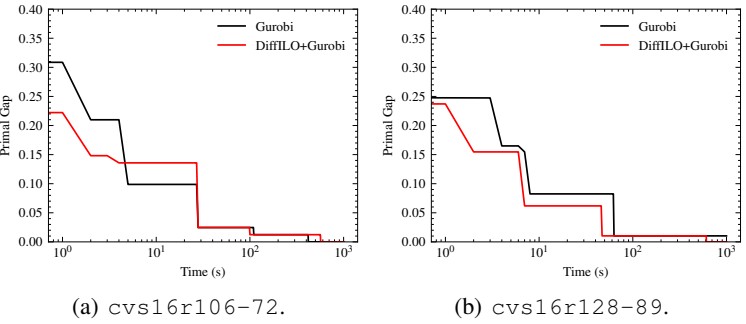

(a) `cvs16r106-72`.        (b) `cvs16r128-89`.

Figure 9: The relative primal gap on the two test instances.

We have conducted an experiment on another subset of MIPLIB. We construct a subset of MIPLIB, called "neos", to validate the effectiveness of DiffILO. Specifically, we collect all binary instances with "neos" in their names and select the instances whose ".mps" files contain no more than 500,000 lines. We identify 25 such instances in total. Then, we randomly select 20 instances for training and use another 5 for test. Among these 5 instances, `neos-952987` and `neos-4382714-ruvuma` derive no feasible solutions solved by Gurobi in 1,000 seconds. We report the solving time on the other 3 test instances, `neos-829552`, neos-831188, `neos18` in Table 7.

Table 7: **Solving time on the 3 test instances from neos.**

|  | neos-829552 | neos-831188 | neos18 |
|---|---|---|---|
| Gurobi | 44.83 | 63.24 | 4.24 |
| Gurobi+DiffILO | 43.08 | 60.91 | 4.28 |

While there were slight overall improvements, they were not significant enough to draw firm conclusions. We attribute this to the inherent heterogeneity of the neos dataset. According to the MIPLIB website, the neos instances originate from diverse scenarios with unknown applications. This poses significant challenges for ML-based approaches, which rely on common patterns and generalizations across instances. Additionally, we find that the heterogeneity among training samples led to unstable training processes, further complicating evaluation. The results demonstrate that training on heterogeneous datasets still pose challenges to DiffILO.

### D.4 TRAINING CURVES

We present the training curves of DiffILO in Figure 10. The consistent progress of training and validation curves shows that DiffILO exhibits good generalization. The consistent progress of losses and objectives shows that DiffILO effectively aligns the training and inference objectives.

### D.5 INFLUENCE OF ADAPTIVE PENALTY COEFFICIENT

We present the training curves with two different settings of the penalty coefficient $\mu$ in Figure 11. The results show that with our proposed adaptive penalty strategy, the training process is robust to different configurations of $\mu$, and the coefficient $\mu$ can adjust itself towards a reasonable value.

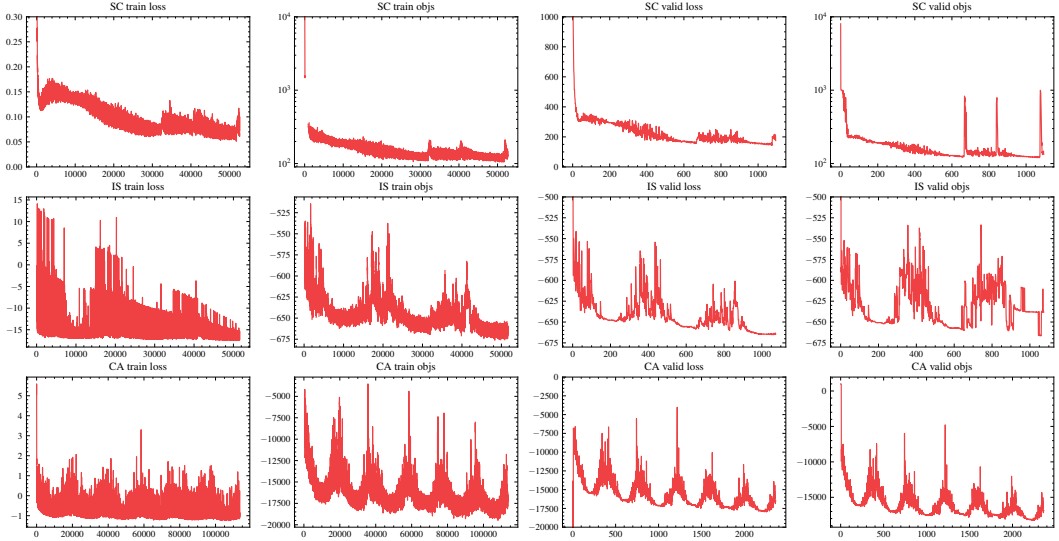

Figure 10: **The learning curves of DiffILO on different datasets.** Objs denotes the objective values of the best-found solutions at each step.

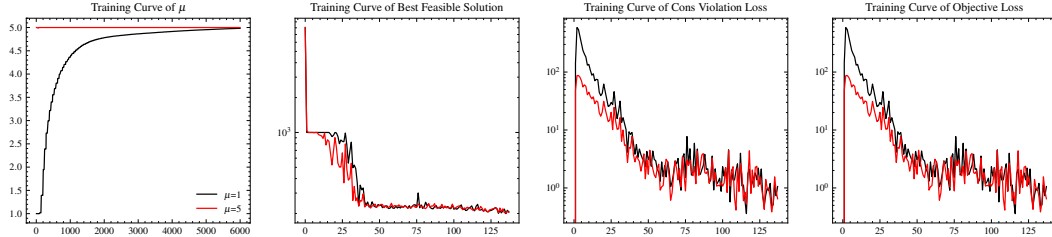

Figure 11: **Training curves for different configurations of the penalty parameter $\mu$.** In the two experiments, we set the initial value of $\mu$ as $1.0$ and $5.0$ (with maximum set as $5.0$), respectively. The parameter $\mu$ is dynamically adaptive during training. Even if we set a small initial value for $\mu$, e.g., $1.0$ here, it can be adaptively adjusted towards a proper value, i.e., $5.0$ here.

## D.6    ABLATION STUDIES

We have conducted experiments to evaluate key choices, and the results are in Figure 12. **The number of samples.** We conduct experiments on a SC dataset, with results shown in Figure 12(a). We evaluated sample sizes of 5, 10, 15, 20, and 25. While larger sample sizes resulted in slightly smoother training curves and smaller sample sizes led to a little early convergence in the early stage of training, the overall results do not show significant differences. This demonstrates the robustness of DiffILO to this parameter. For the main experiments, we just empirically set the sample size to 15. We also compared performance with and without the proposed normalization techniques. The results, presented in FigureD.6(b), show that our normalization method significantly accelerates convergence compared to directly summing all penalty terms. We also tested averaging the constraint penalties instead of summing them, which resulted in worse validation performance and thus is not presented in the figure.

## D.7    GENERALIZATION RESULTS

We further test the zero-shot generalizability of PS and DiffILO. Specifically, the models are trained on small SC instances (with $3,000$ constraints an $2,000$ variables), and tested on large SC instances (with $6,000$ constraints and $4,000$ variables). The results are in Table 8, demonstrating that DiffILO generalizes well to large-sized instances. This may be because the unsupervised training approach

Table 8: **Generalization to large-size datasets.** The models are trained on small SC instances (with 3000 constraints an 2000 variables), and tested on large SC instances (with 6000 constraints and 4000 variables). Results show that DiffILO performs well on large-sized instances, indicating its generalization ability.

|  | SC-small (BKS: 86.45) | | | SC-large (BKS: 79.35) | | |
|---|---|---|---|---|---|---|
|  | 10s | 100s | 1000s | 10s | 100s | 1000s |
| Gurobi | 1031.39 | 87.09 | 86.52 | 993.65 | 85.92 | 79.58 |
| PS+Gurobi | 131.87 | 125.26 | 125.26 | 144.76 | 131.45 | 131.45 |
| DiffILO+Gurobi | **95.65** | **86.78** | **86.48** | **97.83** | **84.72** | **79.55** |

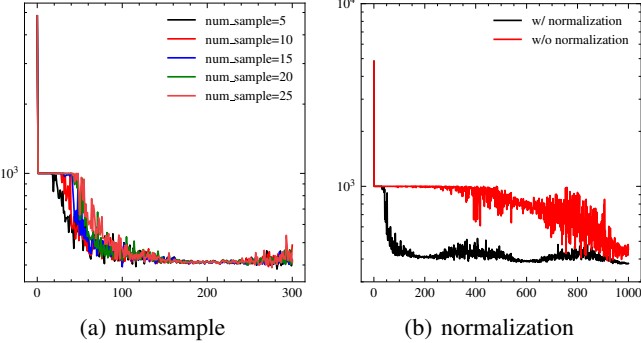

(a) numsample  (b) normalization

Figure 12: Ablation studies on (a) number of samples in each step and (b) the normalization trick used for training.

encourages the model to learn the fundamental mechanisms needed to solve problems, instead of merely memorizing simple statistical patterns in the data, thus outperforming supervised methods.

## D.8 CASE STUDY

We have conducted additional experiments to investigate how the penalty parameter $\mu$ affects the success rate. The results are in Table 9. In this table, we report the number of successes out of 20 trails with different random seeds. The results show that optimizing closed-form merit function is robust against changes in $\mu$. Interestingly, its performance is mainly determined by the random seed. However, it consistently underperforms compared to DiffILO across a broad range of $\mu$ values. With a properly chosen $\mu$, DiffILO achieves a $100\%$ success ratio.

Table 9: The influence of $\mu$ on the success ratio in the case study.

|  | **8** | **9** | **10** | **11** | **12** | **13** | **14** | **15** | **16** | **17** | **18** | **19** | **20** | **25** | **50** | **100** |
|---|---|---|---|---|---|---|---|---|---|---|---|---|---|---|---|---|
| closed-form | 0 | 0 | 0 | 9 | 9 | 9 | 9 | 9 | 9 | 9 | 9 | 9 | 9 | 9 | 9 | 9 |
| DiffILO | 0 | 0 | 0 | 0 | 0 | 2 | 7 | 17 | 20 | 20 | 20 | 20 | 20 | 20 | 19 | 16 |

# E DISCUSSIONS

## E.1 LIMITATIONS

**Sub-optimality** DiffILO is an unsupervised learning approach, which does not learn from the optimal solutions found by traditional solvers, but instead learns to produce solutions itself. There are reasonable concerns that DiffILO may tend to generate sub-optimal solutions, especially when compared with supervised learning approaches. Sub-optimality is essentially a fundamental challenge for most optimization approaches. Therefore, more attention should be paid to addressing this issue and reducing the risk of sub-optimality.

**Research scope** DiffILO mainly focuses on solving integer linear programs (ILPs). Currently, we focus on ILPs with binary variables. More general ILPs with integer variables and mixed-integer linear programs (MILPs) are out of the scope of this paper. This extension poses new challenges, especially in dealing with the unbounded integer and continuous variables. However, we believe that DiffILO provides an avenue for such directions, and we plan to explore the use of differentiable approaches for solving general ILPs and MILPs in future work.

**Waiting for more sophisticated designs** The differentiable ILP solving is still in its early stage and lacks sophisticated designs. For example, the sampled solutions can be stored in a buffer for better sample efficiency. The specific gradient optimization algorithm for such task is not yet developed. Moreover, the current bipartite graph representation and GNN architecture are still simple, and can be further designed in the future.

## E.2 FUTURE AVENUES

**Used for large-scale pretraining** The unsupervised nature of DiffILO makes it suitable for pretraining tasks on large-scale datasets.

**Combination with supervised learning** DiffILO is based on unsupervised learning and may be stuck in sub-optimal. It is promising to combine it with small amounts of supervised data to better overcome the sub-optimality issue.

**Better optimization algorithm** In this work, we use simple Adam for gradient-based optimization. Better and more specificly-used optimizers could be developed in the future. Moreover, integrating traditional methods such as branch-and-bound or large-neighborhood search into our framework could bolster its robustness and help the model navigate complex solution landscapes effectively.

**Extension to more general problems** We note that a contemporaneous ICLR submission (Anonymous, 2024) reports and attempts to tackle the similar limitations. They stated: "Most existing end-to-end machine learning-based methods primarily focus on predicting solutions for binary variables." Their approach involves converting integer variables into binary representations and predicting these binary bits iteratively. This iterative binary prediction approach could be extended to our framework, though it would require additional modifications. We plan to explore this direction in future work. Moreover, While this paper primarily focuses on ILPs, the underlying principles can be extended to non-linear problems. The key lies in the design of the probabilistic model for $\hat{\phi}_j(\hat{\mathbf{x}}) = \mathbb{E}_{\mathbf{x} \sim p(\cdot|\hat{\mathbf{x}})}[\phi_j(\mathbf{x})]$, where $\phi_j(\mathbf{x})$ can be adapted for non-linear constraints. Exploring such extensions is an exciting direction for future work.

**Better model architectures** In the future, better model architectures will be expored to replace the GNN in the current methodology. Some potential practices include soft matching between nodes (Cuturi, 2013; Caron et al., 2021), differentiable clustering for a soft cluster assignment (Stewart et al., 2024), and vector quantization for assigning discrete values (Van Den Oord et al., 2017).

