# OpenReview forum: "Differentiable Integer Linear Programming"
_ICLR.cc/2025/Conference — ICLR 2025 Spotlight_

### Official Review · Reviewer_Rkn8 · 2024-10-23

**Soundness:** 4
**Presentation:** 4
**Contribution:** 3
**Rating:** 8
**Confidence:** 3

**Summary:**

The paper concerns itself with integer linear programs (ILPs), a NP hard optimization problem. Previous works have trained models in a supervised manner to predict near optimal solutions as a heuristic guess to a problem instance. In this work, the authors propose a unsupervised method to train predictors: namely, by using a Bernoulli relaxation of the ILP variable, and reformulating the ILP as a unconstrained problem (via the introduction of a penalty function), a application of the Gumbel-Softmax trick (as a “relaxed Bernoulli”) enables for gradient flow suitable for back -propagation.

The mathematics corresponding to the methodology are clearly presented in detail. The methodology is evaluated empirically on three ILP benchmarks:

- Set covering
- Maximum independent set
- Combinatorial Auctions

and compared to i) traditional solvers ii) Predict-and-search framework, as baselines. In this section the authors also provide practical results e.g. which hyper parameters are crucial, learning rate schedule, which are helpful for practitioners.

**Strengths:**

The paper is well written, with the methodology and experiments both presented in a clear and coherent fashion. As far as I am aware, the unsupervised learning approach is indeed completely novel. Whilst there is a wealth of literature for creating differentiable proxies of CO problems, (for which the paper calls upon multiple tools / results), I believe the overall methodology to be a significant contribution. The presentation of the mathematics underpinning the relaxation and reformulation was particularly well written.

**Weaknesses:**

The methodology in its current form is constrained to using a GNN as a predictor for the Bipartite graph, which seems quite excessive ; the graph structure is simple and GNNs have a high computational complexity (and poor scalability). However, the ideas presented in the work are independent of this and it is nonessential to the method. Below are two suggestions for methods to replace the GNN in the current methodology, both of which would allow for more general architectures (e.g. transformer). These may be worth mentioning as future possible work.

 - Sinkhorn Knop for soft matching between nodes:  see  **[Cuturi et al 2013]** *Sinkhorn distances: Lightspeed computation of optimal transport*. (An example of such an implementation can be seen in **[Caron et al 2021]** *Emerging Properties in Self-Supervised Vision Transformers*)
- Differentiable Clustering for a soft cluster assignment (between a cluster for 0 and 1): see  **[Stewart et al 2023]**  *Differentiable Clustering with Perturbed Spanning Forests*.
- Vector Quantization (not differentiable, but commonly used in practise to assign discrete values): **[van den Oord 2017]** *Neural Discrete Representation Learning*.


As someone who is not familiar with ILPs, it would have been nicer to have further motivation on the real world applications of ILPs, and more intuition as to why DNNs are preferable to predict solutions over other established search methods (please note: I am not questioning either of these points, just pointing out that a more explicit clarification on these would be helpful to a non-expert reader).

**Questions:**

In Remark 5 you mention that you favour the relaxed Bernoulli over using REINFORCE, citing that it does not explicitly propagate the gradients from $\phi_j(x)$. Did you conduct experiments to verify that in practice this is indeed the case? If so this could be interesting to add to the Appendix, (appending a reference to Remark 5).

I believe the following reference would be useful for the paper (regarding smoothing COs): [Berthet 2020] *Learning with Differentiable Perturbed Optimizers*

---

> ### Author Response · Authors · 2024-11-22
> **Response to Reviewer Rkn8 --- Part 1/2**
>
> Dear Reviewer RKn8,
>
> Thank you for your positive and insightful comments. We sincerely hope our rebuttal could adequately address your concerns. If so, we would deeply appreciate it if you could consider raising your score. If not, please let us know your further concerns, and we will continue actively responding to your comments.
>
> ### Weakness 1. Choice of graph structure and GNN.
>
> > The methodology in its current form is constrained to using a GNN as a predictor for the Bipartite graph, which seems quite excessive ; the graph structure is simple and GNNs have a high computational complexity (and poor scalability). However, the ideas presented in the work are independent of this and it is nonessential to the method.
>
> Thank you for your insightful suggestions and for recognizing that our contributions are independent of the GNN choice.
>
> - Using bipartite graphs and GNNs to represent MILP problems is a **widely applied practice** in the field [1], employed in both seminal works [2] and recent advancements [3]. Research on the representation power of such practice has also emerged [4]. However, up to now such practice remains an **advanced and widely used approach** in this domain.
> - We agree there is significant room for improvement in both graph representation and model architecture. We have included related discussions in **Appendix E.2 (Limitation Section)** and plan to explore more efficient representations in the future. Additionally, **unsupervised learning** methods have demonstrated strongly related to **representation learning** and **large-scale pre-training** in many domains such as large language models, computer vision, and drug discovery. We believe our unsupervised approach can serve as **a motivating factor for developing more advanced graph representations and architectures**.
>
> [1] A survey for solving mixed integer programming via machine learning. Neurocomputing, 2023.
>
> [2] Exact combinatorial optimization with graph convolutional neural networks. NeurIPS 2019.
>
> [3] A gnn-guided predict-and-search framework for mixed-integer linear programming. ICLR 2023.
>
> [4] On representing mixed-integer linear programs by graph neural networks. ICLR 2023.
>
> ### Weakness 2. Future works to replace the GNN architectures.
>
> > Below are two suggestions for methods to replace the GNN in the current methodology, both of which would allow for more general architectures (e.g. transformer). These may be worth mentioning as future possible work.
> >
> > - Sinkhorn Knop for soft matching between nodes: see **[Cuturi et al 2013]** *Sinkhorn distances: Lightspeed computation of optimal transport*. (An example of such an implementation can be seen in **[Caron et al 2021]** *Emerging Properties in Self-Supervised Vision Transformers*)
> > - Differentiable Clustering for a soft cluster assignment (between a cluster for 0 and 1): see **[Stewart et al 2023]** *Differentiable Clustering with Perturbed Spanning Forests*.
> > - Vector Quantization (not differentiable, but commonly used in practise to assign discrete values): **[van den Oord 2017]** *Neural Discrete Representation Learning*.
>
> We sincerely thank you for these suggestions, which provide excellent ideas for future exploration. We are already planning to investigate **better representations** and **alternative architectures** to enhance our framework, and your suggestions are greatly appreciated. In response, we have added a new paragraph to **Appendix E.3 (Future Work Section)** to discuss these potential approaches, including the provided references. If our paper is accepted, we welcome you to follow our future works in this direction.
>
> ### Weakness 3. More background introduction.
>
> > As someone who is not familiar with ILPs, it would have been nicer to have further motivation on the real world applications of ILPs, and more intuition as to why DNNs are preferable to predict solutions over other established search methods (please note: I am not questioning either of these points, just pointing out that a more explicit clarification on these would be helpful to a non-expert reader).
>
> Thank you for this helpful suggestion. To address this, we have expanded the **Introduction Section** to include additional details.

---

> ### Author Response · Authors · 2024-11-22
> **Response to Reviewer Rkn8 --- Part 2/2**
>
> ### Question 1. Experiments on REINFORCE method.
>
> > In Remark 5 you mention that you favour the relaxed Bernoulli over using REINFORCE, citing that it does not explicitly propagate the gradients from $\phi_j(x)$. Did you conduct experiments to verify that in practice this is indeed the case? If so this could be interesting to add to the Appendix, (appending a reference to Remark 5).
>
> We **have conducted experiments to demonstrate this claim**, and the details are included in **Appendix D.2**.
>
> - Specifically, we implement a REINFORCE method as a baseline, which computes gradients as
>
>   $$\nabla\_{\hat{x}}\mathbb{E}\_{x\sim p(\cdot|\hat{x})}[C(x)]=\mathbb{E}\_{x\sim p(\cdot|\hat{x})}[C(x)\nabla\_{\hat{x}}\log p(\cdot|\hat{x})],$$
>
>   where $C(x)$ denotes the merit function as defined in (P3). The results show that **REINFORCE fails on this task**, with **all models collapsing towards minimal objectives but significant constraint violations**, even if we set a very large $\mu$.
>
> - As discussed, this failure arises because REINFORCE relies on **random exploration without gradient guidance**. When a solution is reached, the model receives only a reward signal but lacks insight into the components of the reward or the gradient at that point. In such an **extremely high-dimensional search space**, the absence of gradient-directed exploration can lead to convergence on trivial yet infeasible solutions.
> - Per your suggestion, we have also added a **reference to Appendix D.2** in **Remark 5**.
>
> ### Question 2. Reference on smoothing COs.
>
> > I believe the following reference would be useful for the paper (regarding smoothing COs): [Berthet 2020] *Learning with Differentiable Perturbed Optimizers*
>
> Thank you for providing this reference. Berthet et al. (2020) propose a general method to transform discrete optimizers into differentiable operations by perturbing the inputs of a discrete solver with random noise. In our work, we adopt the **Gumbel-Softmax trick** (see **Remark 6**) for reparameterization. According to Berthet et al. (see their **Section 2**), the Gumbel trick can be viewed as a specific example of their perturbed optimizer framework. We have now included this reference in **Remark 6** to acknowledge its relevance.

---

> ### Comment · Reviewer_Rkn8 · 2024-11-22
>
> I thank the author(s) for taking the time to provide such comprehensive responses to the comments, and for having incorporated them via the listed revisions. With this taken into account I have raised my score.

---

> > ### Author Response · Authors · 2024-11-22
> > **Thank you for your kind support.**
> >
> > Dear Reviewer Rkn8,
> >
> > Thanks for your kind support and for helping us improve the paper. We sincerely appreciate your valuable suggestions.
> >
> > Best,
> >
> > Authors

---

### Official Review · Reviewer_NhwL · 2024-11-01

**Soundness:** 3
**Presentation:** 3
**Contribution:** 2
**Rating:** 6
**Confidence:** 3

**Summary:**

This paper introduces Differentiable Integer Linear Programming Optimization (DiffILO), a novel learning method for predicting high-quality Integer Linear Programming (ILP) solutions in an unsupervised manner, without the reliance on traditional solvers. The proposed prediction model is a Graph Neural Network (GNN) module followed by a multilayer perceptron (MLP). By transforming ILPs into a continuous, differentiable, and unconstrained form through probabilistic modeling and the penalty function method, the authors enable the use of gradient descent for optimization. The approach avoids reliance on traditional solvers and labeled data, reducing training time.

**Strengths:**

1. The paper is well-written and clear.
2. Adequate theoretical support is provided for the key steps.
3. As one of the NeurIPS reviewers for this paper, I am pleased to see that the paper includes many of the experimental results requested during the rebuttal period.

**Weaknesses:**

1. Given that there are various relaxations made during the conversion of the ILP to an unconstrained problem, the experiments do not ablate the effect of the choices made at each step. For example, for the relaxation converting the constraint violation into a sampling based objective, it is not clear what the effect of the number of samples is. In the Appendix, the training loss has been modified via some specific form of normalization, but it is not clear what happens to the empirical performance when such normalizations are removed.
2. SC, MIS and CA are easy combinatorial optimization problems and hence identifying feasible solutions without relying on MILP solvers is not challenging. Experiment results on more realistic ILPs (such as those from MIPLIB 2017) should be included in the main paper.

**Questions:**

I am interested in the results and analysis of the MIPLIB experiments. Why were the “neos” datasets chosen for experiments during the NeurIPS rebuttal but not included in the current submission? Instead, the “CVS” datasets were presented. During the NeurIPS rebuttal, after the authors fixed the bugs in the Gurobi configuration, the solving time for Gurobi changed from 1000 seconds to less than 100 seconds.

The experimental results on the neos18 dataset indicate that Gurobi+DiffILO requires a longer solving time than pure Gurobi, and I am curious about the reason for this. I reviewed the problem details of neos18 and the five “CVS” datasets presented in the paper, and found that the number of variables and constraints is smaller in the five “CVS” datasets. For example, neos18 has 11,402 constraints, while the five “CVS” datasets have fewer than 5,000 constraints. Does this suggest that DiffILO may not perform well on more complex benchmarks? Could you provide the experimental results on the “neos” datasets and explain why Gurobi+DiffILO performs worse than Gurobi?

---

> ### Author Response · Authors · 2024-11-22
> **Response to Reviewer NhwL --- Part 1/2**
>
> Dear Reviewer NhwL,
>
> Thank you for your insightful and valuable comments. We sincerely hope our rebuttal could adequately address your concerns. If so, we would deeply appreciate it if you could consider raising your score. If not, please let us know your further concerns, and we will continue actively responding to your comments.
>
> ### Weakness 1. Ablation studies.
>
> > Given that there are various relaxations made during the conversion of the ILP to an unconstrained problem, the experiments do not ablate the effect of the choices made at each step. For example, for the relaxation converting the constraint violation into a sampling based objective, it is not clear what the effect of the number of samples is. In the Appendix, the training loss has been modified via some specific form of normalization, but it is not clear what happens to the empirical performance when such normalizations are removed.
>
> Thank you for your constructive suggestions. We have carefully designed and validated our key methodological choices through comparative experiments. To address your concerns, we have added additional analyses in our experiments to evaluate key choices.
>
> - **The number of samples.** First, in the **case study in the main text**, we investigated the effect of sample size on a toy example. We observed that increasing the number of samples led to more stable convergence in the early training stages but had no significant impact on the final results. To provide more general evidence, we added experiments on a SC dataset, with results shown in **Appendix D.5, Figure 12 (a)**. We evaluated **sample sizes of 5, 10, 15, 20, and 25**. While larger sample sizes resulted in slightly smoother training curves and smaller sample sizes led to a little early convergence in the early stage of training, the overall results **do not show significant differences**. This demonstrates **the robustness of DiffILO** to this parameter. For the main experiments, we just empirically set the sample size to 15.
> - **The normalization.** We also compared performance with and without the proposed normalization techniques. The results, presented in **Appendix D.5, Figure 12 (b)**, show that our normalization method **significantly accelerates convergence compared to directly summing all penalty terms**. We also tested averaging the constraint penalties instead of summing them, which resulted in worse validation performance.
>
> Notice that in **Appendix C.2**, we present three useful training tricks:
>
> - **Normalization** (ablation study in **Appendix D.5**),
> - **Adaptive penalty coefficient $\mu$** (ablation study in **Appendix D.5**), and
> - **Learning rate annealing** (observe the training curves in **Figure 10**).
>
> Each of these techniques has now been empirically validated.
>
> ### Weakness 2. Experiments on MIPLIB 2017 included in the main paper.
>
> > SC, MIS and CA are easy combinatorial optimization problems and hence identifying feasible solutions without relying on MILP solvers is not challenging. Experiment results on more realistic ILPs (such as those from MIPLIB 2017) should be included in the main paper.
>
> Thank you for your valuable suggestion. We have revised the paper to include **MIPLIB 2017** results in **Section 4 (Experiment Section)** in the main paper.

---

> ### Author Response · Authors · 2024-11-22
> **Response to Reviewer NhwL --- Part 2/2**
>
> ### Question. Results on the CVS and neos datasets.
>
> > I am interested in the results and analysis of the MIPLIB experiments. Why were the “neos” datasets chosen for experiments during the NeurIPS rebuttal but not included in the current submission? Instead, the “CVS” datasets were presented. During the NeurIPS rebuttal, after the authors fixed the bugs in the Gurobi configuration, the solving time for Gurobi changed from 1000 seconds to less than 100 seconds.
> >
> > The experimental results on the neos18 dataset indicate that Gurobi+DiffILO requires a longer solving time than pure Gurobi, and I am curious about the reason for this. I reviewed the problem details of neos18 and the five “CVS” datasets presented in the paper, and found that the number of variables and constraints is smaller in the five “CVS” datasets. For example, neos18 has 11,402 constraints, while the five “CVS” datasets have fewer than 5,000 constraints. Does this suggest that DiffILO may not perform well on more complex benchmarks? Could you provide the experimental results on the “neos” datasets and explain why Gurobi+DiffILO performs worse than Gurobi?
>
> Thank you for these detailed observations and questions. Below, we provide a thorough explanation.
>
> - During the NeurIPS rebuttal, we included experiments on the **neos dataset**, and here we quote the results (solving time) reported during NeurIPS rebuttal:
>
>   |                | neos-829552 | neos-831188 | neos18 |
>   | -------------- | ----------- | ----------- | ------ |
>   | Gurobi         | 44.83       | 63.24       | 4.24   |
>   | Gurobi+DiffILO | 43.08       | 60.91       | 4.28   |
>
>   While there were slight overall improvements, they were **not significant enough to draw firm conclusions**. We attribute this to the inherent **heterogeneity** of the neos dataset. According to [the MIPLIB website](https://miplib.zib.de/instance_details_neos18.html), the neos instances originate from diverse scenarios with **unknown applications**. This poses significant challenges for ML-based approaches, which **rely on common patterns and generalizations across instances**. Additionally, we find that the heterogeneity among training samples led to **unstable training processes**, further complicating evaluation.
>
> - In this revision, we use the **CVS datasets** to demonstrate the effectiveness of DiffILO. Specifically, the **CVS dataset** consists of **homogeneous instances** (Capacitated Vertex Separator problems), all derived from the same application domain. These instances exhibit more consistent patterns, aligning better with the assumptions of ML-based approaches.
>
> - Although CVS instances have fewer constraints (under 5,000), they are **more challenging** than neos18. Gurobi fails to solve most CVS instances within 1,000 seconds, while it solves neos18 in seconds. Moreover, benchmarks like SC, IS, and CA are also difficult, as many instances remain unsolved by Gurobi within 1,000 seconds. Therefore, **we cannot simply conclude that DiffILO underperforms on complex benchmarks**.
>
> - To provide a complete picture, we have also included the **results and analysis on the neos dataset** in **Appendix D.3**. We also incorporate discussions about the training on heterogeneous datasets in **Appendix E.2 (Limitation Section)**.

---

> > ### Comment · Reviewer_y4sA · 2024-11-22
> >
> > It would be great to include the neos results in the main paper explaining the limitation on heterogeneous instances.

---

> > > ### Author Response · Authors · 2024-11-23
> > > **Thanks for your suggestion and we have included the results in the main paper.**
> > >
> > > Dear Reviewer y4sA,
> > >
> > > We sincerely thank you for your thoughtful and constructive suggestions. We have revised the layout and included the neos results along with the corresponding analysis in **Section 4 in the main paper**. We hope these additions provide greater clarity and enhance the reader's understanding of this line of research. Please let us know if you have further concerns or suggestions, and we will continue actively responding to your comments.
> > >
> > > Best,
> > >
> > > Authors

---

> > ### Comment · Reviewer_NhwL · 2024-11-25
> >
> > I would like to thank the authors for their detailed responses to my review comments. I have reviewed the authors' responses to all reviewers and the updated submission. Their explanations have addressed my concerns and clarified the reasons for their experimental settings as well as the advantages of their proposal. I appreciate the effort made to address the concerns raised and will be increasing my score.

---

> > > ### Author Response · Authors · 2024-11-25
> > > **Thank you for your kind support.**
> > >
> > > Dear Reviewer NhwL,
> > >
> > > Thanks for your kind support and for helping us improve the paper. We sincerely appreciate your valuable suggestions.
> > >
> > > Best,
> > >
> > > Authors

---

### Official Review · Reviewer_LD6k · 2024-11-02

**Soundness:** 3
**Presentation:** 4
**Contribution:** 4
**Rating:** 8
**Confidence:** 4

**Summary:**

The authors proposes DiffILO, a new approach that uses machine learning to solve integer linear programs (ILPs) without supervision and without traditional solvers. DiffILO transforms ILPs into continuous, differentiable, and unconstrained problems through probabilistic modeling and applied the penalty based merit function, allowing for optimization using gradient descent directly. That is, there is no need in calling solver at all. Instead, the model (which predict solution to ILP) is trained via backpropagating the merit function.

Unlike supervised methods that require labeled data typically obtained by solving ILPs, DiffILO operates in an unsupervised manner, which reduces training time. The approach has been tested on small-to-medium scaled ILP datasets, demonstrating its ability to speed up the training process and produce feasible solutions. These solutions may differ from those generated by supervised methods.

**Strengths:**

Overall, I think this is an interesting perspective into learning predictive models for obtaining approximation solution for combinatorial problems. Majority of previous approaches use a solver calls in some way to learn that predictive mapping, whereas here it is done by defining a differentiable (a.e.) function that serves as an objective to optimize. In this regard, I find it similar to the decision-focused learning (DFL) or predict-then-optimize framework [1,2,3] where the task is to learn a model which maps observable features into latent representation (e.g. coefficients in LP objective) used by solvers. Here, the training formulation is similar but the solution is predicted instead of latent representation. Particularly [3] draws this connection between these two domains and apply it for MINLPs. I encourage authors to add this line of research and elaborate on this. Other strengths of the paper include:

- theoretical justification of the continuous relaxation applied for this problem. Although ILP covers a lot of important class of problems, however I don't see these to be directly extended into non-linear case.
- experimental results look convincing in terms of both runtime and solution quality. Although adding larger scale experiments would be beneficial;
- the method is intuitive to understand and makes sense to me.
- can be directly applied to speed up the runtime for traditional solvers;


[1] A. N. Elmachtoub and P. Grigas. Smart “predict, then optimize”. arXiv:1710.08005

[2] A. Ferber, B. Wilder, B. Dilkina, and M. Tambe. MIPaaL: Mixed integer program as a layer.

[3] A. Zharmagambetov, B. Amos, A. Ferber, T. Huang, B. Dilkina, and Y. Tian (2023): "Landscape Surrogate: Learning Decision Losses for Mathematical Optimization Under Partial Information".

**Weaknesses:**

Some are mentioned in Strengths above. Additionally, I think that the supervised approaches a bit underperforming here due to limited sample size. With enough data for supervision, I think those approaches should also improve drastically, especially for larger scale problems.

**Questions:**

- typo in line 198;

---

> ### Author Response · Authors · 2024-11-22
> **Response to Reviewer LD6k**
>
> Dear Reviewer LD6k,
>
> Thank you for your positive and insightful comments. We sincerely hope our rebuttal could adequately address your concerns. If so, we would deeply appreciate it if you could consider raising your score. If not, please let us know your further concerns, and we will continue actively responding to your comments.
>
> ### Weakness1. Related work.
>
> > In this regard, I find it similar to the decision-focused learning (DFL) or predict-then-optimize framework [1,2,3] where the task is to learn a model which maps observable features into latent representation (e.g. coefficients in LP objective) used by solvers. Here, the training formulation is similar but the solution is predicted instead of latent representation. Particularly [3] draws this connection between these two domains and apply it for MINLPs. I encourage authors to add this line of research and elaborate on this.
> >
> > [1] A. N. Elmachtoub and P. Grigas. Smart “predict, then optimize”. arXiv:1710.08005
> >
> > [2] A. Ferber, B. Wilder, B. Dilkina, and M. Tambe. MIPaaL: Mixed integer program as a layer.
> >
> > [3] A. Zharmagambetov, B. Amos, A. Ferber, T. Huang, B. Dilkina, and Y. Tian (2023): "Landscape Surrogate: Learning Decision Losses for Mathematical Optimization Under Partial Information".
>
> Thank you for your constructive suggestion. We have incorporated these references into **Section 2 (Related Work)** and elaborated on this line of research.
>
> ### Weakness2. Extension to non-linear cases.
>
> > Although ILP covers a lot of important class of problems, however I don't see these to be directly extended into non-linear case.
>
> We appreciate your thoughtful feedback. While this paper primarily focuses on ILPs, the underlying principles can be extended to non-linear problems. The key lies in the design of the probabilistic model for $\hat{\phi}_j(\hat{\mathbf{x}})=\mathbb{E}\_{\mathbf{x}\sim p(\cdot|\hat{\mathbf{x}})}[\phi_j(\mathbf{x})]$, where $\phi_j(\mathbf{x})$ can be adapted for non-linear constraints. Exploring such extensions is an exciting direction for future work, and we have added this discussion to **Appendix E.3 (Future Work Section).**
>
> > experimental results look convincing in terms of both runtime and solution quality. Although adding larger scale experiments would be beneficial;
>
> Thank you for this suggestion. We have conducted experiments to evaluate generalization to larger-scale instances in **Table 8 (Appendix D.7)**. We plan to conduct more comprehensive evaluations on larger datasets in future work.
>
> ### Weakness 3. Limited sample size.
>
> > Additionally, I think that the supervised approaches a bit underperforming here due to limited sample size. With enough data for supervision, I think those approaches should also improve drastically, especially for larger scale problems.
>
> - Thank you for raising this point. To evaluate this, we **conducted additional experiments** on SC by **doubling the training dataset size for PS**, increasing it to 480 training instances and 120 validation instances. The results are presented below.
>
>   |          | SC (BKS: 86.45) |        |        |
>   | -------- | --------------- | ------ | ------ |
>   |          | 10s             | 100s   | 1000s  |
>   | PS (300) | 131.87          | 125.26 | 125.26 |
>   | PS (600) | 134.55          | 122.48 | 122.48 |
>   | DiffILO  | 95.65           | 86.78  | 86.48  |
>
>   Interestingly, increasing the dataset size did not lead to significant improvements for PS, suggesting that the current dataset size may already be sufficient for this approach. The training dataset we used aligns with common practices in prior work and represents a reasonable balance between data size and computational cost.
>
> - Notably, the **scaling law** (where larger datasets lead to significant performance gains) for supervised learning in the ILP domain has not yet been well-established. The performance can be affected by factors beyond data size, such as **model expressiveness** and the **predictive paradigm**. While we believe that supervised methods could benefit from further innovations in these areas, our primary goal in this work is to demonstrate the feasibility of **unsupervised learning** as a complementary and alternative approach. We believe future developments---such as combining unsupervised and supervised techniques---will leverage the strengths of both frameworks and lead to further performance improvements.
> - Additionally, supervised methods rely on solver-generated labels, which make **larger datasets more costly and time-consuming** to generate. In contrast, our unsupervised approach provides a significant advantage by **reducing training time** while maintaining strong performance.
>
> ### Question. Typo.
>
> > typo in line 198;
>
> Thank you for pointing out this, and we have revised the typo here.

---

> ### Author Response · Authors · 2024-11-26
> **We are looking forward to your feedback.**
>
> Dear Reviewer LD6k,
>
> We are writing as the authors of the paper titled "Differentiable Integer Linear Programming" (ID: 7722). We sincerely thank you for your time and efforts during the rebuttal process. We are looking forward to your feedback to understand if our responses have adequately addressed your concerns. If so, **we would deeply appreciate it if you could consider raising your score**. If not, please let us know your further concerns, and we will continue actively responding to your comments. We sincerely thank you once more for your insightful comments and kind support.
>
> Best,
>
> Authors

---

> ### Author Response · Authors · 2024-11-29
> **We are looking forward to your feedback.**
>
> Dear Reviewer LD6k,
>
> We are writing as the authors of the paper titled "Differentiable Integer Linear Programming" (ID: 7722). We sincerely thank you for your time and efforts during the rebuttal process. Since the discussion phase will be ending soon, we are looking forward to your feedback to make sure that our responses have adequately addressed your concerns. If so, **we would deeply appreciate it if you could consider raising your score**. If not, please let us know your further concerns, and we will continue actively responding to your comments.
>
> Best,
>
> Authors

---

> > ### Comment · Reviewer_LD6k · 2024-12-02
> >
> > Thanks for the reply. I have no further concerns and keep my original score.

---

> > > ### Author Response · Authors · 2024-12-04
> > > **Thank you for your kind support.**
> > >
> > > Dear Reviewer LD6k,
> > >
> > > Thanks for your kind support and for helping us improve the paper. We sincerely appreciate your valuable suggestions.
> > >
> > > Best,
> > >
> > > Authors

---

### Official Review · Reviewer_y4sA · 2024-11-04

**Soundness:** 4
**Presentation:** 4
**Contribution:** 4
**Rating:** 8
**Confidence:** 4

**Summary:**

The authors propose an interesting approach for unsupervised learning in ILP. Evaluating it in several binary programming settings, and investigating the approach itself empirically in various ways. The approach itself relies on considering that a model predicts a continuous solution where each entry represents the probability of assigning a given decision variable to 1 or 0. The model is then trained to optimize a loss that combines the expected objective value with the expected constraint violation. The expected constraint violation is estimated by sampling several solutions and computing expected constraint violation using the samples. The benefit of the unsupervised approach is that it bypasses the need to expensively collect solutions from training instances. Additionally, the authors propose that the unsupervised approach helps improve predictive performance by encouraging the predicted objects to represent feasible solutions. The authors present theoretical motivations for the approach, as well as thorough empirical evaluation on toy examples to give insights as to how the approach works.

Overall, the work is interesting while there is some room for improvement, if the authors address my comments I am eager to increase my score.

**Strengths:**

The strengths of the approach are that it doesn’t require expensive optimization solving for training time. Most of the literature hasn’t considered this as it is assumed that practitioners are willing to spend time upfront training a model that can be deployed on many instances, but nevertheless it can be impactful in some settings to require less training time, for instance it can be possible to train on many more problem instances given the same training time, or even larger instances given that training instances don’t need to be solved to optimality.

The approach itself is well motivated and the paper is well-written.

The illustrative toy example is helpful for giving intuition for how the approach works in a simple setting.

**Weaknesses:**

The approach is proposed for general ILP; however, the approach seems to be tailored to binary programs. There is a remark stating that ILP can be reduced to binary programs; however, it would help strengthen the paper if there were experimental results validating that this approach can be used in general ILP tasks to make that claim (such as on MIPLIB instances other than the CVS dataset), or to rephrase the method as working for binary programs.

Theorem 2 statement 2: it seems that this direction of solvability/optimality doesn’t really apply in this setting since the predicted continuous x is always fractional as considered below in the approach. Is there any indication that the distribution being optimal for P2 has any implication about the optimality wrt P1 of the discrete solutions that the distribution represents? Is there any indication of whether the probability distribution puts weight on suboptimal solutions?

It is unclear whether the approach would outperform baselines other than the single PS baseline considered here as more recent work with available code seems to have outperformed the predict and search approach such as the two cited works. However, it would be interesting to see if the unsupervised approach could be integrated in the settings considered in previous work as well.

Specific comments:
-	Remark 2 ends in “Otherwise,” is something missing there?
-	Figure 4 is missing
-	Toour is missing a space

**Questions:**

How different are the initial solutions compared to the solutions after one round of neighborhood search? (i.e. after solving the optimiazation problem with constraint (9) added?

Why are the Zheng 2024 and Huan 2024 baselines not included as they seemed to surpass the PS approach and provide implementations.

How many decision variables do the different settings have? It is somewhat unclear why this method would be more robust to changes in delta than baseline approaches. Is it the case that the predicted solution is already close to optimal, so a large neighborhood doesn’t need to be searched?

How does the approach generalize to different kinds of problems? Either to larger instances or out of distribution instances e.g. MIPLIB?

What are the feasibility rates for PS? They are given for DiffILO but not present for the baseline. It seems figure 4 is missing.

How is mu determined? Is it determined as a hyperparameter? Or adaptively selected to ensure feasibility?

---

> ### Author Response · Authors · 2024-11-22
> **Response to Reviewer y4sA --- Part 1/4**
>
> Dear Reviewer y4sA,
>
> Thank you for your positive and insightful comments. We sincerely hope our rebuttal could adequately address your concerns. If so, we would deeply appreciate it if you could consider raising your score. If not, please let us know your further concerns, and we will continue actively responding to your comments.
>
> ### Weakness1. Extension to general ILPs.
>
> > The approach is proposed for general ILP; however, the approach seems to be tailored to binary programs. There is a remark stating that ILP can be reduced to binary programs; however, it would help strengthen the paper if there were experimental results validating that this approach can be used in general ILP tasks to make that claim (such as on MIPLIB instances other than the CVS dataset), or to rephrase the method as working for binary programs.
>
> Thank you for pointing this out. While we agree that validating the approach on general ILPs would indeed strengthen the paper, implementing this within the rebuttal timeframe is non-trivial. Below, we address your concerns in detail.
>
> - As noted in **Remark 1**, **bounded ILPs can theoretically be converted into binary forms**. This **ensures theoretical completeness**. However, directly applying this transformation increases the number of variables, leading to inefficiencies in both representation and computation. Additionally, our review of existing works shows that **most state-of-the-art end-to-end solving methods**, such as NeuralDiving [1], PS [2], and ConPaS [3], also **focus on binary variables**. Additionally, most commonly used benchmarks in this domain contain only binary variables. We have revised **Remark 1** to explicitly clarify the statement, and included further discussions in **Section E.2 (Limitation Section)**.
>
> - We note that [a contemporaneous ICLR submission](https://openreview.net/forum?id=scdGzuwC9u) [4] reports and attempts to tackle the similar limitations. They stated:
>   *"Most existing end-to-end machine learning-based methods primarily focus on predicting solutions for binary variables."* Their approach involves converting integer variables into binary representations and predicting these binary bits iteratively. This iterative binary prediction approach could be extended to our framework, though it would require additional modifications. We plan to explore this direction in future work. For now, included additional discussions in **Section E.3 (Future Work Section)** to discuss on the potential extensions.
>
> [1] Solving mixed integer programs using neural  networks.
>
> [2] A gnn-guided predict-and-search framework for mixed-integer linear programming. ICLR 2023.
>
> [3] Contrastive predict-and-search for mixed integer linear programs. ICML 2024.
>
> [4] A Reoptimization Framework for Mixed Integer Linear Programming with Dynamic Parameters. ICLR 2025 submission.
>
> ### Weakness2. Theorem 2.
>
> > Theorem 2 statement 2: it seems that this direction of solvability/optimality doesn’t really apply in this setting since the predicted continuous x is always fractional as considered below in the approach. Is there any indication that the distribution being optimal for P2 has any implication about the optimality wrt P1 of the discrete solutions that the distribution represents? Is there any indication of whether the probability distribution puts weight on suboptimal solutions?
>
> Thank you for your detailed inquiry. Let us revisit **Theorem 2 Statement 2**, which provides two key insights:
>
> 1. If we find an optimal solution  $\hat{x}^*$ to (P2), many of its components (determined by $\mathcal{I}_c$) will be binary rather than fractional.
> 2. An optimal solution $x^*$ to (P1) can be derived by setting $x_i^*=\hat{x}_i^*$ for binary $\hat{x}_i^*$, and choosing either $0$ or $1$ for the remaining components without affecting the solution's optimality.
>
> Loosely speaking, this means that an optimal solution to (P2) closely aligns with an optimal solution to (P1). For the **binary components** in $\hat{x}^*$, their values must **match those in** $x^*$ for (P1). For the **fractional components**, however, their specific values do not impact the optimality, as either $0$ or $1$ would suffice **without affecting the optimality**.
>
> Intuitively, the optimal state for (P2) corresponds to **a probability distribution** that assigns weight **exclusively to the optimal solutions of (P1)**. As the optimization of (P2) progresses, this distribution $\hat{x}^*$ will theoretically converge toward a deterministic binary solution concentrated on the optimal values of (P1). Although the actual predicted outputs by our model are initially **fractional** (indicating **uncertainty in the distribution**), the optimization of (P2) guides these predictions toward a **binary** solution (indicating a **deterministic distribution** concentrated on the optimal solutions of (P1)).

---

> ### Author Response · Authors · 2024-11-22
> **Response to Reviewer y4sA --- Part 2/4**
>
> ### Weakness3 & Question 2. Additional baselines.
>
> > It is unclear whether the approach would outperform baselines other than the single PS baseline considered here as more recent work with available code seems to have outperformed the predict and search approach such as the two cited works. However, it would be interesting to see if the unsupervised approach could be integrated in the settings considered in previous work as well.
>
> > Why are the Zheng 2024 and Huan 2024 baselines not included as they seemed to surpass the PS approach and provide implementations.
>
> Thank you for your valuable suggestions. To address your concerns, we have included **additional baselines**, **ConPaS** (Huan 2024) [1] and **DDIM** (Zheng 2024) [2].
>
> - **Implementation Details.** Since ConPaS [1] does not provide publicly released code, we implement the approach based on the paper's details. For DDIM [2], we used the authors' released code. These baselines were used to generate solutions for the SC instances, and the results are included in **Appendix D.2**.
> - **Comparison Results.** The results show that ConPaS (Huan 2024) still fails to generate feasible solutions across most instances. DDIM (Zheng 2024) demonstrates strong feasibility rates and successfully generates feasible solutions for all instances. However, when considering solution quality, **DiffILO still outperformed DDIM** in terms of objective values. This highlights the strength of DiffILO in producing higher-quality solutions.
> - It is important to note that both ConPaS and DDIM are built upon **advanced supervised learning techniques**, including **contrastive learning** and **diffusion models**. These methods represent the culmination of much development in supervised learning paradigms. In contrast, DiffILO pioneers a new line of research by introducing an **unsupervised learning framework**. We believe that integrating more advanced techniques into this framework will further improve DiffILO’s performance in the future.
>
> [1] Contrastive predict-and-search for mixed integer linear programs. ICML 2024.
>
> [2] Effective Generation of Feasible Solutions for Integer Programming via Guided Diffusion. KDD 2024.
>
> ### Weakness 4 & Question 5. Specific comments
>
> > Specific comments: - Remark 2 ends in “Otherwise,” is something missing there? - Figure 4 is missing - Toour is missing a space
>
> > What are the feasibility rates for PS? They are given for DiffILO but not present for the baseline. It seems figure 4 is missing.
>
> - **Remark 2**: Thank you for pointing out the typo and we have corrected it.
> - **Figure 4**: The figure is located at the top of Page 8. Hyperlinks in the main text can direct readers to the appropriate figures. Figure 4 compares the objective values of solutions generated by different methods. The results indicate that **PS fails to produce feasible solutions without solver assistance** (i.e., feasibility rate = 0). When augmented with solver heuristics, PS can generate feasible solutions; however, **DiffILO consistently outperforms PS in solution quality**.
> - **Typo**: Thank you for pointing out this typo, and we have revised it.

---

> ### Author Response · Authors · 2024-11-22
> **Response to Reviewer y4sA --- Part 3/4**
>
> ### Question1 & Question3.
>
> > How different are the initial solutions compared to the solutions after one round of neighborhood search? (i.e. after solving the optimization problem with constraint (9) added?
>
> > How many decision variables do the different settings have? It is somewhat unclear why this method would be more robust to changes in delta than baseline approaches. Is it the case that the predicted solution is already close to optimal, so a large neighborhood doesn’t need to be searched?
>
> - Thanks for your question. We have tested the differences between the initial solutions $\mathbf{x}_0$ and the solutions after one round of neighborhood search $\mathbf{x}_1$ (i.e., solving the optimization problem with constraint (9) added). Specifically, the initial solution $\mathbf{x}_0$ was generated by DiffILO and used as the starting point. Gurobi was then employed to refine the solution under the constraint (9), with a time limit of 100 seconds. For each instance, we calculated the difference $\|\mathbf{x}_0-\mathbf{x}_1\|_1$. Across $10$ SC instances, the average difference was **25.8**, indicating that our generated solutions are already **very close to the refined solutions** found via the neighborhood search algorithm.
>
> - In our search algorithm, we **only have one hyperparameter** $\Delta$, which controls the radius of the neighborhood search. This is a very easy setting and does not require extensive hyperparameter tuning. Setting $\Delta = 200$ restricts the search for 200 variables among all variables that differ from the initial solution. PS adopts a more complex search algorithm, with three hyperparameters: $k_0$, $k_1$, and $\Delta$. Here $k_0$ denotes the number of fixed $0$'s, and $k_1$ denotes the number of fixed $1$'s in the solution. The parameter $\Delta$ then represents the number of changes allowed in the $k_0+k_1$ variables. These multiple hyperparameters in PS significantly influence the final results and require careful tuning. However, tuning three interdependent hyperparameters can be **challenging and computationally expensive**. Our approach, with a simpler hyperparameter setting, demonstrates comparable or better performance while requiring fewer decision variables.
> - The aforementioned results indeed suggest that our model generates **predicted solutions close to optimal solutions**, reducing the need for extensive neighborhood search. This may arises because DiffILO has been trained to solve these instances itself rather than from supervisions, thus leading to better robustness. In contrast, PS relies on the average of solutions as predictive labels during supervised training. This averaging leads to **blurred labels and inaccurate predictions**, which represent an aggregate rather than a sharp approximation of optimal solutions. Consequently, PS requires more intensive neighborhood search to refine its predictions effectively.
>
> ### Question4. Generalization.
>
> > How does the approach generalize to different kinds of problems? Either to larger instances or out of distribution instances e.g. MIPLIB?
>
> Thanks for your suggestion. We have conducted experiments to demonstrate the generalization ability of DiffILO, and the details are in **Appendix D.7**. Specifically, the models are trained on small SC instances (with $3,000$ constraints an $2,000$ variables), and tested on large SC instances (with $6,000$ constraints and $4,000$ variables). For your convenience, we quote the results below.
>
> |                | SC (3000, 2000, BKS: 86.45) |        |        | SC (6000, 4000, BKS: 79.35) |        |        |
> | -------------- | --------------------------- | ------ | ------ | --------------------------- | ------ | ------ |
> |                | 10                          | 100    | 1000   | 10                          | 100    | 1000   |
> | Gurobi         | 1031.39                     | 87.09  | 86.52  | 993.65                      | 85.92  | 79.58  |
> | Gurobi+PS      | 131.87                      | 125.26 | 125.26 | 144.76                      | 131.45 | 131.45 |
> | Gurobi+DiffILO | 95.65                       | 86.78  | 86.48  | 97.83                       | 84.7   | 79.55  |
>
> The results demonstrate that DiffILO generalizes well to large-sized instances. This may be because the unsupervised training approach encourages the model to learn the fundamental mechanisms needed to solve problems, instead of merely memorizing simple statistical patterns in the data, thus outperforming supervised methods.

---

> ### Author Response · Authors · 2024-11-22
> **Response to Reviewer y4sA --- Part 4/4**
>
> ### Question6. Penalty coefficient $\mu$
>
> > How is mu determined? Is it determined as a hyperparameter? Or adaptively selected to ensure feasibility?
>
> The penalty coefficient μ\muμ is dynamically and adaptively determined, inspired by the adaptive temperature approach in soft actor-critic algorithms. Details are provided in **Appendix C.2**. Specifically, after each epoch, we update the coefficient $\mu$ according to the updating rule
> $$
> \mu_{k+1} = \mu_{k} + \text{mu\\_step} * (\text{cons} - \text{cons\\_targ}),
> $$
> where $\text{cons}$ denotes the average constraint violation in this epoch, and $\text{cons}\_\text{targ}$ is the target value of the average constraint violation. Empirically the hyperparameter $\text{mu}\_\text{targ}$ is set as no more than $1$ (according to the range of coefficients), as this indicates that there exist solutions with no constraint violation in a probabilistic sense. This dynamic way for tuning $\mu$ can effectively improve the algorithm robustness against the choice of $\mu$. We present the training curves with different values the parameter $\mu$ and analyze the influence of the adaptive strategy for $\mu$ in **Appendix D.5**.

---

> ### Author Response · Authors · 2024-11-26
> **We are looking forward to your feedback.**
>
> Dear Reviewer y4sA,
>
> We are writing as the authors of the paper titled "Differentiable Integer Linear Programming" (ID: 7722). We sincerely thank you for your time and efforts during the rebuttal process. We are looking forward to your feedback to understand if our responses have adequately addressed your concerns. If so, **we would deeply appreciate it if you could consider raising your score**. If not, please let us know your further concerns, and we will continue actively responding to your comments. We sincerely thank you once more for your insightful comments and kind support.
>
> Best,
>
> Authors

---

> > ### Comment · Reviewer_y4sA · 2024-11-26
> >
> > I thank the authors for their responses, paper clarifications, and thorough investigation of the approach in various settings. The method gives promising results that generalize well, and the authors give experimental indication of the need for future work on solving heterogeneous problem instances. These clarifications and experiments have addressed my concerns and I  raise my score.

---

> > > ### Author Response · Authors · 2024-11-29
> > > **Thank you for your kind support.**
> > >
> > > Dear Reviewer y4sA,
> > >
> > > Thanks for your kind support and for helping us improve the paper. We sincerely appreciate your valuable suggestions.
> > >
> > > Best,
> > >
> > > Authors

---

### Official Review · Reviewer_FT2F · 2024-11-04

**Soundness:** 2
**Presentation:** 3
**Contribution:** 4
**Rating:** 6
**Confidence:** 4

**Summary:**

This paper proposes a new learn-to-optimize paradigm that trains a solution predictor without relying on traditional solvers to generate label data. As a result, the entire pipeline is significantly faster by avoiding solver runs. The paradigm is based on designing a Lagrangian loss for the predicted solution and iteratively updating the predictor using the gradient of the Lagrangian loss.

**Strengths:**

The idea of replacing solvers in the training pipeline is intriguing. Indeed, I can envision many problem classes where off-the-shelf solvers may underperform compared to simple gradient-descent-based algorithms. The proposed method could be highly effective for such problems.

**Weaknesses:**

When comparing their results to existing methods based on solver-generated labels, the authors overlook an important limitation of their approach: their unsupervised learning method does not learn from optimal ILP solutions and may instead be trained to only produce significantly sub-optimal solutions.

Gradient descent algorithms for MILP problems are not new (e.g., see the paper "Feasibility Jump: an LP-free Lagrangian MIP heuristic") and they generally converge to a suboptimal, heuristic solution. By performing gradient descent on the Lagrangian loss, the unsupervised learning method proposed in this paper essentially learns from heuristic solutions, which may fall far short of optimality.

To ensure a fair comparison, I believe the authors should modify the solver-based supervised learning pipelines by setting limits on (i) the solving time and (ii) the number of branch-and-bound nodes. Most off-the-shelf solvers can find a good solution in a short time, with the extended solving time largely dedicated to ensuring optimality. Since the authors are not learning from optimal solutions, they should compare their approach to existing methods without optimality requirements.

**Questions:**

see weakness.

---

> ### Author Response · Authors · 2024-11-22
> **Response to Reviewer FT2F --- Part 1/3**
>
> Dear Reviewer FT2F,
>
> Thank you for your insightful and valuable comments. We sincerely hope our rebuttal could adequately address your concerns. If so, we would deeply appreciate it if you could consider raising your score. If not, please let us know your further concerns, and we will continue actively responding to your comments.
>
> ### Weakness1. Sub-optimality of gradient descent algorithms.
>
> > When comparing their results to existing methods based on solver-generated labels, the authors overlook an important limitation of their approach: their unsupervised learning method does not learn from optimal ILP solutions and may instead be trained to only produce significantly sub-optimal solutions.
> >
> > Gradient descent algorithms for MILP problems are not new (e.g., see the paper "Feasibility Jump: an LP-free Lagrangian MIP heuristic") and they generally converge to a suboptimal, heuristic solution. By performing gradient descent on the Lagrangian loss, the unsupervised learning method proposed in this paper essentially learns from heuristic solutions, which may fall far short of optimality.
>
> We appreciate your concern regarding the potentially sub-optimal solutions generated by unsupervised approaches, as they do not explicitly learn from optimal solutions. Sub-optimality is indeed a fundamental challenge faced by most optimization algorithms. We have included a new paragraph in **Appendix E.2 (i.e., Limitation Section)** discussing this issue in detail. Below, we respond to your concern point-by-point.
>
> **1. Our unsupervised learning approach demonstrates comparable and even better results than supervised learning.**
>
> While sub-optimality remains a challenge for unsupervised learning, we have observed that it achieves competitive or even superior performance compared to supervised learning approaches. Here we analyze the reasons.
>
> - **Supervised learning methods also face sub-optimality issues.** Existing supervised learning approaches rely on solver-generated labels, which are the average of solutions obtained by running solvers like Gurobi for 3,600 seconds. However, solvers typically return also sub-optimal rather than globally optimal solutions under such time constraints, as confirmed by our experiments. This means that the labels used for supervised learning are also sub-optimal.
> - **DiffILO achieves better alignment between training and inference objectives.** In supervised learning, the training objective is to minimize prediction error and learn from the solution distribution. However, the solution distribution often represents an average of possible solutions. Sampling from such a distribution does not necessarily generate high-quality feasible solutions. In contrast, our unsupervised method trains the model by directly evaluating solution quality during inference, resulting in a better alignment between training and inference goals.
> - **Unsupervised learning fosters deeper understanding.** While supervised learning simplifies training by providing explicit labels, it often leads to models capturing only superficial statistical patterns. In contrast, our unsupervised approach requires the model to independently discover solutions, encouraging a more intrinsic understanding of the optimization problem. This is analogous to **a student who excels through independent problem-solving outperforming another who relies heavily on guidance from teachers**.
>
> Thus, while sub-optimality is a shared challenge across both paradigms, and experiments show that our approach performs competitively and even better. Still, we want to emphasize that our goal is not to prove that unsupervised learning is better than supervised learning. Instead, we aim to offer a novel framework for unsupervised learning in this domain, which demonstrates promising potential.
>
> **2. Additional significant advantages of unsupervised learning.**
>
> Each method has its own advantages and disadvantages. Despite the risk of local optima, unsupervised learning offers several distinct advantages over supervised learning.
>
> - **Unsupervised learning approaches do not rely on solver-generated labels.** By eliminating the need for solver-generated labels, unsupervised learning drastically reduces training time while achieving comparable or better results.
> - **Foundation for large-scale pre-training.** In many fields, such as large language models, computer vision, and drug discovery, unsupervised learning has proven essential and fundamental for **large-scale pre-training** and for **developing foundation models**. (Empirically, the key factors for foundation models are: unsupervised learning approach, scalable models, and large datasets.) Our work represents the first step toward this in the ILP domain. Although in its early stages, we expect it to lay the groundwork for potential large-scale pre-training.

---

> ### Author Response · Authors · 2024-11-22
> **Response to Reviewer FT2F --- Part 2/3**
>
> **3. As a trained predictor, not an exact solver, our approach can still find its application scenarios.**
>
> Notice that DiffILO is designed as a predictor rather than an exact solver. As stated in the reference paper [1], even though the produced solutions are not always optimal, the its ability to provide high-quality heuristic solutions rapidly makes it valuable in many applications.
>
> - **Quick Generation of High-Quality Feasible Solutions.** DiffILO excels in generating high-quality feasible solutions swiftly, often outperforming supervised learning approaches in terms of feasibility and quality. In real-world applications such as real-time planning, where decision-making speed is crucial, **a feasible solution obtained quickly is often more practical than waiting for an optimal one**. DiffILO meets this need by producing solutions in minimal time.
> - **Enhancing traditional solvers.** DiffILO does not aim to replace traditional solvers but to enhance them. By providing a strong initial solution, it accelerates the solver's convergence to optimal or near-optimal solutions. Starting from the feasible solution generated by DiffILO, solvers can often explore the solution space more effectively and achieve better results in less time.
>
> **4. Our efforts to alleviate sub-optimality.**
>
> We have taken substantial steps to address the challenge of sub-optimality and mitigate the risk of local optima in our approach.
>
> - **Novel Optimization Framework.** We have developed a **probabilistic modeling framework** combined with a **sampling-based penalty function** to reduce the likelihood of local optima and enhance solution quality. Preliminary results from our **case study** demonstrate the effectiveness of this method in mitigating sub-optimality.
>
> - **Future Directions to Address Sub-optimality.** We plan to explore additional strategies to tackle sub-optimality more comprehensively. For instance:
>
>   - **Hybrid Training Approaches:** Combining **unsupervised learning** with **small amounts of supervised data**, as seen in other domains, could further improve model performance.
>   - **Incorporating Optimization Techniques:** Integrating traditional methods such as **branch-and-bound** or **large-neighborhood search** into our framework could bolster its robustness and help the model navigate complex solution landscapes effectively.
>
>   We have included these potential directions in **Appendix E.3 (Future Work Section)**.

---

> ### Author Response · Authors · 2024-11-22
> **Response to Reviewer FT2F --- Part 3/3**
>
> ### Weakness2. Fairness of comparison.
>
> > To ensure a fair comparison, I believe the authors should modify the solver-based supervised learning pipelines by setting limits on (i) the solving time and (ii) the number of branch-and-bound nodes. Most off-the-shelf solvers can find a good solution in a short time, with the extended solving time largely dedicated to ensuring optimality. Since the authors are not learning from optimal solutions, they should compare their approach to existing methods without optimality requirements.
>
> Thank you for your insightful suggestion regarding fairness in comparisons. We fully appreciate your concern and have taken this into our experiments. Below, we provide a detailed response to clarify our approach.
>
> **1. The solver settings have been properly configured.**
>
> - We appreciate your advice for fair comparison. However, to our best knowledge, setting limits on solving time or the number of branch-and-bound nodes influences the stopping condition rather than the solving process itself. **Therefore, such settings would not accelerate solving**.
> - We assume that you advise us to ensure that the solvers **prioritize finding feasible solutions rather than focusing on proving optimality**. If so, indeed, in our experiments, **we have configured solver settings** for this purpose. Specifically, for Gurobi, we used the ["MIPFocus" parameter](https://www.gurobi.com/documentation/10.0/refman/mipfocus.html) and set  `m.Params.MIPFocus = 1`. For SCIP, we used the  ["AGGRESSIVE" parameter](https://listserv.zib.de/pipermail/scip/2021-February/004217.html) and set `m.setHeuristics(SCIP_PARAMSETTING.AGGRESSIVE)`. These settings instruct the solvers to prioritize finding feasible solutions quickly rather than proving optimality. These configurations, detailed in **Appendix C.3**, align with widely accepted practices in the PS paper [2]. Additionally, when generating training labels for supervised learning baselines, we applied the same settings to ensure high-quality labels.
>
> **2. The comparison is fair, with the same solver configurations.**
>
> We want to clarify that our approach is not designed to replace traditional solvers but to enhance them by **providing high-quality initial heuristic solutions**. These predictive solutions help solvers accelerate their optimization process. Importantly, both experiments---with and without our approach---use **the same solver configurations**, ensuring a fair comparison. Below, we further elaborate the results.
>
> - In **Figure 4**, we compare the **solutions generated by our method** to the **heuristic solutions** generated by solvers (i.e., the heuristic process after pre-solving but before the root node). This heuristic process focuses on **quickly finding feasible solutions**, typically within seconds, **without attempting to prove optimality**.
> - In **Table 1** and **Figure 5**, we report the objective values achieved by different methods **at 10, 100, and 1,000 seconds**, as well as the full solving curves. The results show that across different time horizons---both short and long---our approach outperforms the baselines in terms of solution quality. We believe this aligns with the type of comparison you suggested.
>
> [1] Feasibility Jump: an LP-free Lagrangian MIP heuristic. Mathematical Programming Computation.
>
> [2] A GNN-Guided Predict-and-Search Framework for Mixed-Integer Linear Programming. ICLR 2023.

---

> > ### Comment · Reviewer_FT2F · 2024-11-27
> >
> > Thank you for your detailed response; most of my concerns have been addressed. I would like to offer an additional comment on the configuration of solvers. The parameter m.Params.MIPFocus you used primarily influences the branch-and-bound tree size. While focusing on feasibility might seem beneficial, it doesn't necessarily lead to finding a solution more quickly. Often, concentrating on improving the dual bound can reduce the search space more effectively and lead to faster solutions. In addition to early stopping the solver by limiting time or node counts, you might consider using parameters such as SolutionLimit.

---

> ### Author Response · Authors · 2024-11-26
> **We are looking forward to your feedback.**
>
> Dear Reviewer FT2F,
>
> We are writing as the authors of the paper titled "Differentiable Integer Linear Programming" (ID: 7722). We sincerely thank you for your time and efforts during the rebuttal process. We are looking forward to your feedback to understand if our responses have adequately addressed your concerns. If so, **we would deeply appreciate it if you could consider raising your score**. If not, please let us know your further concerns, and we will continue actively responding to your comments. We sincerely thank you once more for your insightful comments and kind support.
>
> Best,
>
> Authors

---

> ### Author Response · Authors · 2024-11-28
> **Thank you for your kind support and reponse to your further comments.**
>
> Dear Reviewer FT2F,
>
> Thank you for your kind support and constructive suggestions. Below, we address your further comments regarding the configuration of solvers.
>
> ### 1. The parameter `m.Params.MIPFocus`
>
> According to the [Gurobi document](https://docs.gurobi.com/projects/optimizer/en/10.0/reference/parameters.html#parametermipfocus), this parameter influences the solver's high-level solution strategy as follows.
>
> - `MIPFocus=0`: Strikes a balance between finding new feasible solutions and proving that the current solution is optimal.
> - `MIPFocus=1`: Prioritizes finding feasible solutions quickly.
> - `MIPFocus=2`: Focuses more attention on proving optimality.
> - `MIPFocus=3`: Focuses on optimizing the objective bound, particularly useful when the best objective bound is moving very slowly (or not at all).
>
> To assess its impact, we tested the results using `m.Params.MIPFocus` values of $0$, $1$, $2$, and $3$ on the **SC** dataset. We report the objective values at different time intervals ($10$s, $100$s, $1000$s) in the following table.
>
> |                                 | 10s     | 100s  | 1000s |
> | ------------------------------- | ------- | ----- | ----- |
> | Gurobi (`MIPFocus=0`)           | 1031.39 | 88.16 | 86.78 |
> | Gurobi (`MIPFocus=1`)           | 1031.39 | 87.09 | 86.52 |
> | Gurobi (`MIPFocus=2`)           | 1031.39 | 89.41 | 86.87 |
> | Gurobi (`MIPFocus=3`)           | 1031.39 | 88.36 | 87.05 |
> | DiffILO + Gurobi (`MIPFocus=1`) | 95.65   | 86.78 | 86.48 |
>
> As shown, **DiffILO + Gurobi consistently outperforms Gurobi with different `MIPFocus` settings** in terms of objective values,
>
> ### 2. Early stopping by limiting time or node counts
>
> In our experiments, we report the objective values at $10$s, $100$s, and $1000$s in **Table 1** and provide a graphical representation of the solving progress in **Figure 5**. Notably, applying a time or node limit equivalently corresponds to early stopping, and the objective values at $10$s are the same as the output of stopping the solver early using `m.Params.TimeLimit=10`. These results demonstrate that, even with limited solving time, **DiffILO can significantly improve the solutions obtained by the solver**.
>
> ### 3. Early stopping by `m.Params.SolutionLimit`
>
> According to the [Gurobi document](https://docs.gurobi.com/projects/optimizer/en/10.0/reference/numericcodes/statuscodes.html#tablestatuscodes), this parameter limits the number of feasible solutions found before stopping. When we set `m.Params.SolutionLimit = 1`, Gurobi terminates upon finding the first feasible solution, regardless of its quality.
>
> We conduct **additional experiments** with `m.Params.SolutionLimit=1`. Our results show that when this parameter is set, Gurobi stops as soon as it finds a feasible solution. This leads to **identical objective values** compared to the heuristic solutions (which we report in **Figure 4**) found by the default Gurobi solver. Specifically, we report the average objective values obtained by different methods on **SC** in the following table. Here, Gurobi (Heuristic) indicates the default heuristic mode in Gurobi, which is called before exact solving and used to find initial heuristic solutions. We obtained these results by extracting the foudn heuristic solutions from the Gurobi logging files. These solutions are quickly found within seconds.
>
> |                            | Obj     |
> | -------------------------- | ------- |
> | Gurobi (Heuristic)           | 2404.16 |
> | Gurobi (SolutionLimit=1)   | 2404.16 |
> | DiffILO                    | 159.37  |
> | DiffILO + Gurobi (Heuristic) | 96.03   |
>
> The results still show that **DiffILO outperforms both Gurobi with the default settings and Gurobi with early stopping** in terms of finding initial heuristic solutions with better objective values.

---

### Public Comment · ~Do_Hoang_Khoi_Nguyen1 · 2025-04-08

Dear authors, thank you very much for your excellent work. I truly enjoyed studying your paper. However, I could not find the source code at the GitHub link provided in the manuscript. May I kindly ask if you could share the code for your work? It would be greatly helpful for further understanding and potential application. Looking forward to your response, and thank you in advance.

---

> ### Public Comment · ~Zijie_Geng1 · 2025-04-09
> **Code Now Available**
>
> Dear Dr. Do Hoang Khoi Nguyen,
>
> Thank you very much for your interest in our work. Sorry for the delay as I was working on organizing the code for release. The latest version of our code is now available at: [https://github.com/MIRALab-USTC/L2O-DiffILO](https://github.com/MIRALab-USTC/L2O-DiffILO).
> Please feel free to reach out if you have any questions or feedback.
>
> Best regards,
>
> Zijie Geng

---

> > ### Public Comment · ~Youval_Kashuv1 · 2025-04-11
> > **Missing Data**
> >
> > Dear authors,
> >
> > Thank you for uploading your code. Would it be possible to also provide the datasets used in your experiments? This would greatly facilitate reproduction of your results. Thank you.

---

### Public Comment · ~Youval_Kashuv1 · 2025-04-14
**Missing Data**

Dear authors,

Thank you for making your code available at the GitHub repository. I've been exploring the code base, but I'm having difficulty locating the datasets used in your experiments.

Would it be possible to either:
1. Share the datasets directly, or
2. Provide detailed instructions on the expected data format?

I would like to understand the data structure so I can format my own data appropriately to use with your implementation.

Thank you for your consideration.

---

> ### Public Comment · ~Zijie_Geng1 · 2025-04-16
> **Dataset Shared**
>
> Dear Dr. Youval Kashuv,
>
> Thank you for your interest and valuable feedback. I have shared our example dataset through the Google Drive link: [https://drive.google.com/drive/folders/1G9icDM_UVld8tY9tabU4WFGYJq_vtYk5?usp=sharing](https://drive.google.com/drive/folders/1G9icDM_UVld8tY9tabU4WFGYJq_vtYk5?usp=sharing). Instructions regarding the project structure as well as the data format can be found in our github repository: [https://github.com/MIRALab-USTC/L2O-DiffILO](https://github.com/MIRALab-USTC/L2O-DiffILO), so you can also follow the instructions there to run DiffILO on your own datasets. Please feel free to reach out or open a GitHub issue if you have any further questions.
>
> Best regards,
>
> Zijie Geng

---

### Meta-Review · Area_Chair_76Sc · 2024-12-20

**Metareview:**

This paper introduces DiffILO, a novel method for solving Integer Linear Programs (ILPs). It relies on a probabilistic modeling approach to transform ILPs into unconstrained, differentiable problems, enabling gradient descent optimization. Unlike supervised methods, DiffILO operates unsupervised, reducing training time. Tests on small-to-medium ILPs demonstrate its ability to accelerate training and produce feasible solutions. The reviewers are enthusiastic about this work, and recommend to accept.

Personally, I think that this work is interesting, but would strengthen the "related works" section, since the idea of gradient-based approaches for LPs, and probabilistic relaxations are not new, as pointed out by some of the reviewers.

**Additional Comments On Reviewer Discussion:**

The reviewers were enthusiastic about this work, there was some discussion with the authors.

---

### Decision · Program_Chairs · 2025-01-22

Accept (Spotlight)